# Hierarchical secure key assignment scheme

**Baris Celiktas**[1]*, **Ibrahim Çelikbilek**[2], **Sueda Guzey**[2], **Enver Ozdemir**[2]

**1** Isik University, Computer Engineering Department, Istanbul, Türkiye, **2** Istanbul Technical University, Informatics Institute, Istanbul, Türkiye

☉ These authors contributed equally to this work.
* baris.celiktas@isikun.edu.tr

## Abstract

This work presents a novel hierarchical key assignment mechanism for access control, designed to be computationally lightweight and optimized for digital environments with structured access policies. By leveraging orthogonal projection and distributing a basis to each group, it enables flexible and efficient left-to-right and top-down access structures. The scheme ensures that parent groups can derive the secret keys of their child groups while preventing unauthorized reverse access. It is resilient against collusion attacks and privilege escalation, offering robust key recovery and indistinguishability properties. Moreover, it guarantees strong key indistinguishability under adversarial models and facilitates a secure rekeying process without reliance on a trusted third party. To demonstrate practical efficiency, we provide a full analytical complexity evaluation showing that key derivation requires at most $\mathcal{O}(n_i^2)$ operations, where $n_i$ is the dimension of the assigned subspace. For typical deployment parameters used in the experiments, the total key material per user remains compact ($\approx$ 3,072 bits), significantly smaller than well-known post-quantum schemes such as Dilithium-5 (38,912 bits). The storage requirement scales linearly with the number of groups ($ck+1$ bases for $c$ groups with at most $k$ members), ensuring that even large hierarchies remain lightweight. Our evaluation further shows that selective rekeying affects only the descendants of the modified group, resulting in communication overhead of $\mathcal{O}(m'\lambda)$ bits, where $m'$ is the number of affected users and $\lambda$ is the key length. These results collectively highlight the scheme's scalability, low storage footprint, and suitability for large access hierarchies.

## Introduction

In the digital realm, data confidentiality is ensured through cryptographic primitives such as symmetric keys or public-key algorithms. Encrypted data can be restored to its original form only by using the secret key utilized for the encryption function. The access control policy for this key should align with the data controller's policy, which may involve intricate rules and structures, especially in complex organizational environments. Extracting the key usually requires the approval of multiple entities, and

**Data availability statement:** All relevant data are within the manuscript and its Supporting information files. https://figshare.com/articles/software/Source_Code_for_the_Hierarchical_Key_Assignment_Scheme_HKAS_/30822809?file=60188627.

**Funding:** The author(s) received no specific funding for this work.

**Competing interests:** The authors have declared that no competing interests exist.

a secret sharing algorithm can effectively manage the desired key access control in some cases, as outlined in [1]. However, an access control policy requiring approvals from multiple users, each with a unique classification level assigned by the data controller, may pose practical challenges and limitations when implemented using a secret sharing algorithm. Numerous studies on hierarchical key assignment schemes have been published since Akl and Taylor introduced their hierarchical access control scheme in 1983 [2]. However, our motivation for conducting this research is rooted in the fact that unresolved issues still need to be addressed to develop a practical access mechanism for hierarchical structures.

Various organizations handling mission-critical data are embracing the current trend of transitioning their services to digital platforms. In such scenarios, organizations can choose public or private cloud services to facilitate data flow and adapt access policies to fit the digital environment. However, the use of public clouds raises significant concerns, particularly in sectors like military, healthcare, and banking, where confidentiality and privacy are paramount. Beyond general concerns regarding confidentiality, availability, integrity, reliability, data lock-in, and regulatory compliance, integrating the data controller's access policy into a digital medium remains a complex and ongoing challenge within the research community [3,4]. Due to the aforementioned concerns, many organizations have been postponing their digitalization plans despite the public cloud deployment model provides many advantages, especially in total time-cost [5].

This study introduces a practical key access policy that securely mirrors the hierarchical access policy of data controllers. By implementing the resulting scheme, concerns regarding the transfer of private data to the public cloud may be mitigated. The proposed scheme introduces a flexible hierarchical key assignment protocol founded on inner product spaces, leveraging the mathematical framework of orthogonal projection (OP). The foundational concepts related to this scheme were previously outlined in a doctoral dissertation [6].

This study centers on the Bell-LaPadula (BLP) hierarchical multilevel lattice-based model, designed to ensure confidentiality within hierarchical organizational structures, particularly in contexts such as government and military organizations. One of the primary objectives of the model is to ensure simple security, where groups with lower classification privileges are unable to access or read objects of a higher classification level. Therefore, the read-down model is adopted for hierarchical access control. Fig 1 depicts a multilevel hierarchical organizational structure, where the number of levels corresponds to the classification tiers established by the data controller.

Many organizations employ hierarchical management mechanisms within their organizational structures. These mechanisms are often established by dividing the organization into functional areas or smaller organizational units/groups, represented as $G_i$. The data controller defines a security classification level for each $G_i$ in order to access the key/data. It is assumed that only members with higher classification levels can access data of lower classification levels if a parent-child relationship exists.

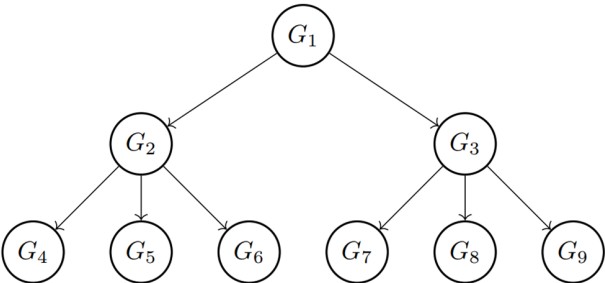

**Fig 1. A left-to-right and top-down policy-based partially ordered set (poset) within the hierarchical structure.**

Since access control is managed by the data controller and maintained on-premises, a cloud storage provider, acting as a service entity, must not gain access to any mission-critical user data. To address this, a scheme is required to enable the secure utilization of public cloud services for diverse purposes. This study presents the initial stage of such a scheme, focusing on resolving the key access control challenges faced by users.

In the subsequent sections, we utilize directed acyclic graphs (DAG), referred to as access graphs, and represented by $G(V,E)$, where $V$ is a finite set of classes (vertices) and $E$ is a set of paired vertices (edges). Let $V$ consists of sets $\{V_1, V_2, \ldots, V_n\}$. For example, the graph $G(9,8)$ in Fig 1 is a DAG with nine vertices and eight directed edges. The symbol $\leq$ denotes a partial order (binary relation - antisymmetric, transitive, and reflexive) on $V$. Consequently, $(V, \leq)$ forms a partially ordered set (poset). The security classes $V_i$, or equivalently the groups $G_i$s, are disjoint. Let $x, y \in V$, where $x \leq y$ indicates two disjoint classes. This relation implies that users in class $x$ and it means that the users in class $x$ can access all the data accessible to users in class $y$. In other words, $G_x \leq G_y$ signifies that any user in $G_x$ can access data belonging to $G_y$, and $G_x$ has a classification level higher than or equal to $G_y$. Additionally, $G_x$ must be one of the parent classes of $G_y$. Figs 1 and 2 depict the key access structure of the proposed scheme.

- The root parent $G_1$ has two children ($G_2$, $G_3$).
- Each of these children serves as a parent to other classes:
  - Parent $G_2$ has three children ($G_4$, $G_5$, $G_6$).
  - Parent $G_3$ also has three children ($G_7$, $G_8$, $G_9$).

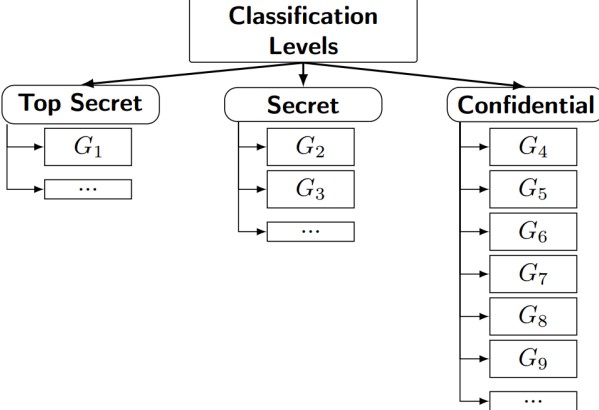

**Fig 2. The classification levels based on taxonomic ranks.**

Each child has only one immediate parent. All parents, except for the root parent $G_1$, also have an immediate parent as well as children. The secret key for each child class can be derived by its parent classes and by the members of that child class. Even if two classes have the same classification level (as shown in Fig 2), if they do not share a parent-child relationship as illustrated in Fig 1, the key of one class cannot be derived by the other. This ensures the integrity and security of the hierarchical access control model.

The application of inner product spaces to the design of key assignment schemes can be traced back to the foundational study [7], in which orthogonal projection methods were introduced within a non-hierarchical framework. In that work, projection-based techniques served a dual purpose: they enabled both authentication and key establishment to be achieved simultaneously, thereby demonstrating the utility of linear algebraic constructs in cryptographic protocol design. Building upon this foundation, the extension of inner product spaces to hierarchical key establishment was first systematically investigated in the corresponding author's doctoral dissertation. That dissertation provided the initial theoretical framework for adapting projection methods to hierarchical structures, thereby laying the groundwork for more complex access control scenarios.

The present study advances this line of inquiry by refining and modifying the dissertation's approach, offering what is, to the best of our knowledge, the first explicit and pioneering application of inner product spaces to hierarchical key establishment. This contribution represents a significant departure from earlier non-hierarchical treatments, as it demonstrates how the algebraic properties of inner product spaces can be harnessed to enforce hierarchical relationships among keys in a mathematically rigorous manner. Subsequent research has further developed this trajectory by analyzing projection-based hierarchical key assignment schemes (HKAS) and comparing with other well-known methods. These later works have shown that projection techniques can be systematically integrated into hierarchical access control, thereby confirming the broader applicability of inner product space methods in cryptographic key management [8].

The structure of this paper is as follows: The *Literature review section* presents a comprehensive literature review on key assignment schemes for hierarchical access control. The *Preliminaries section* outlines the foundational concepts underlying our proposed scheme, including inner product spaces, Gram-Schmidt (GS) method, and orthogonal projection (OP). The *Proposed scheme section* details the proposed scheme, covering its core principles, system initialization, key derivation, and dynamic update processes. The *Experimental results section* showcases the experimental results, including performance evaluations, security analyses, and comparative studies. Finally, The *Conclusion and future work section* offers a concise summary and conclusion.

## Literature review

This section provides a review of the literature on key assignment and access control schemes specifically designed for hierarchical infrastructures.

Several key assignment schemes have been proposed for hierarchical structures, with many of them being based on the Akl and Taylor scheme, [2,9–16] and they are based on a partially ordered set (poset) hierarchy. These schemes do not include features for granting temporary access to users and also overlook the fact that key updates are based on poset hierarchy, which can result in high costs for large hierarchical structures. Furthermore, the work [17] analyzes the Akl-Taylor scheme using the definitions proposed by [18]. It also introduces a method for selecting public parameters that ensure a secure variation of the scheme, resistant to key recovery attacks under the RSA assumption.

The study discusses using a Lucas function-based time-bound property to enhance time and performance efficiency and address key update challenges [19]. These time-bound related schemes can be classified into two categories: the first relies on tamper-proofed machines [20] and the second [19,21] is based on public information. However, tamper-proof machines are costly, unsuitable for cloud environments, and diminish user convenience, whereas public information can be efficiently leveraged due to the cloud's capability to ensure broad network access. The secret key of lower classification levels is derived using a combination of public information and the user's own key. Consequently, the amount of

public information is a critical factor in evaluating the efficiency of the key assignment scheme. It should be noted that the schemes presented in [19] are not designed to provide access to users for a certain period of time.

The subsequent proposals for hierarchical key assignment schemes all aim to achieve a secure, dynamic, and cost-effective approach. The effectiveness and security of these schemes can be evaluated based on several critical parameters, including:

1. **The amount of public and private information required:** The terms "private and public storage" refer to the amount of information that is allocated to each class in the hierarchical structure for the derivation of secret keys. The allocation of private information to classes is carried out by the data controller to restrict access to only the users of the desired classes. Conversely, the data controller distributes all public information to all classes in accordance with the predefined hierarchical structure.

2. **Key derivation complexity:** This refers to the computational cost or number of operations required for key derivation, which should be kept minimal and manageable.

3. **Key update complexity/cost due to the changes in the hierarchical structure:** This metric pertains to the cost required for updating keys and it should be minimized and manageable. A scheme should enable addition or removal of classes without requiring re-distribution of any private information.

4. **Collusion (aka collaborative/key recovery-$KR$) attack resistance:** A secure hierarchical key assignment scheme should ensure that the derivation of the secret key of any class is protected against collusion of users in its child classes. This property is referred to key recovery security ($KR_s$) in the literature.

5. **The key indistinguishability secure ($KI_s$):** The secret key should be indistinguishable from a random string of the same length. A scheme that satisfies this property is called $KI_s$ and it also satisfies key recovery security, $KR_s$. Note that while a scheme is $KI_s$ then it is also $KR_s$, the converse is not necessarily true, as stated in [21].

6. **Resistance to privilege creep problem:** A hierarchical key assignment scheme must ensure that if the classification level of a member $U$ or its membership is changed, the member cannot use their previous privileges during the transition period. This means that the scheme should provide both forward and backward secrecy.

The proposed key management scheme in [2] aims to address the problem of data sharing and multi-group management in hierarchical structures. Users in a communication or computer system are classified into disjoint sets $G_1, G_2, G_3, ..., G_n$, where the relationship between the classes forms a poset hierarchy. Each group is assigned a security level, and users in a group can access data stored by users in the same or lower security level groups, but not in higher level groups. Although this approach is useful for distributed and secure systems, it has limitations. For example, it does not fully address multi-level security problems, and it cannot be adapted dynamically to the security policy set by the data controller. Another issue is that users can use keys permanently at higher security levels, which requires regular replacement and redistribution of keys and incurs significant storage and communication costs. Moreover, the key derivation is expensive, and the scheme is only $KR_s$ according to [21]. With a large number of users, this scheme becomes cost-ineffective due to the large number of keys kept by each user [9]. Additionally, storing public information for each security class requires significant storage [11,16].

The work presented in [9] provides an improved scheme of [2], also known as a canonical assignment, aimed to reduce the storage requirement for public information necessary to control data access within a user group ordered hierarchically, especially if the number of classes is significantly large [19]. However, the storage need is not completely eliminated [16]. Additionally, any user belonging to a group can access the data of lower-level groups since they can derive the keys of lower-level group users, and the scheme is only $KI_s$.

The hierarchical scheme based on symmetric-key cryptography proposed in [10] utilizes a tree structure in which security classes are organized as a rooted tree, exemplifying a partially ordered set (poset) hierarchy. The scheme employs

an iterative method based on a one-way function, allowing for efficient computation of input images while making it computationally infeasible to determine pre-images. Using this approach, the key for a child class can be securely derived. A notable innovation of the scheme is its ability to insert new security classes without requiring changes to the keys of existing classes. This is in contrast to previous schemes, such as [2,9], where the insertion of a new security class required changing all keys not associated with the new class, resulting in a substantial burden, particularly for distributed and large infrastructures. Additionally, the key derivation process does not rely on any additional public parameters, simplifying its implementation. However, the scheme has notable drawbacks: the computational overhead during key derivation can become significant, especially when deriving keys for the lowest class that must be generated by the root of the tree. Moreover, the scheme's efficiency is limited to organizations with fewer than eleven security levels.

The work presented in [11] proposed a method similar to [2], but it employs a bottom-up key derivation policy. Unlike other approaches such as [2,9], this method allows new security classes to be added without altering the keys of existing classes. Furthermore, the public storage requirements for classes are significantly lower than those in [2,9]. Additionally, the scheme demonstrates improved memory efficiency compared to [2].

The works [12,13] proposed schemes utilizing Newton's method and one-way functions. However, these schemes are highly time-intensive due to the significant computation time required for key generation and derivation. Furthermore, they are vulnerable to collaboration attacks, as highlighted in [14].

As proposed by [15], an optimal heuristic algorithm for key distribution in a tree hierarchy uses an up-bottom design approach. This method efficiently generates and derives keys while minimizing the storage requirements for general parameters. However, similar to the approach in [10], its applicability is limited to tree hierarchies.

In contrast, [16] modified the algorithm proposed in [15] to support poset hierarchies. In this scheme, a user in a parent class can derive the keys for users in child classes using a one-way function and their own cryptographic key. However, the reverse is not permitted, as users in a child class are not authorized to derive the key of a parent class. This design ensures that the scheme satisfies the $KR_s$ property.

The approach proposed in [19] employs a time-bound mechanism, derived from [2], to restrict the continuous usage of keys by higher-level users in class $C$. This scheme allows membership in $C$ for a limited time period. For instance, a user in $C_x$ can derive $K_y$ from $K_x$ at a specific time $t$ only if $C_y \leq C_x$ and $t_1 \leq t \leq t_2$, where $t_1$ and $t_2$ define the valid time interval.

Therefore, encrypted data can only be held by a user for a limited time period. The hierarchical structure in this approach has an optimal bandwidth for broadcasting data to authorized users. Each user can only receive data that has been granted access, and unauthorized users cannot obtain data by listening to the broadcast. Moreover, a user of the parent class can grant privileges to another user to decrypt encrypted data, providing flexibility. Unlike previously proposed key assignment schemes based on poset hierarchy, this approach is not dependent on the number of classes in the hierarchy. However, since users must hold keys to access authorized data for a limited time period, it is not computationally efficient as expected. Although it requires less communication and storage cost, it is computationally inefficient due to the need for costly public key computation and costly computations that occur during implementation [20]. Additionally, it is not $KI_s$ if at least three users collaborate then they can access the keys [22].

The work presented in [20] is an improvement over the time-bound key assignment scheme proposed in [19] inspired by [2]. It addresses the performance and implementation issues of the previous scheme by assigning different keys to each user in the hierarchical structure. The scheme employs a tamper-resistant machine that performs simple operations and is not accessible to the controller, ensuring a high level of security. A Trusted Agent (*TA*) and a secure one-way hash function $h$ are also used in the scheme. The key generation process is economically unfeasible to derive from the public value, and users in child classes cannot derive the keys of users in parent classes, making it $KR_s$. The scheme ensures that no user can derive any key beyond their authorized time period. Performance analysis shows that the proposed scheme is much more efficient than the previous scheme presented in [19]. The scheme requires a low-cost, tamper-resistant machine with small storage and computational complexity. However, the scheme is vulnerable to collusion attacks by three or more users, making it not $KR_s$ [22,23].

In [24], various key assignment schemes found in the literature are classified into five categories. Trivial Key Assignment Scheme (TKAS) is simple, but its key update/change process is difficult. Trivial Key Encrypting Key Assignment Scheme (TKEKAS), on the other hand, has an easier key update process and is useful if the key is compromised. In Direct Key Encrypting Key Assignment Scheme (DKEKAS), public data is large, but private storage is minimal, and the key update process is easier. Node-Based Key Assignment Scheme (NBKAS) requires only a single secret value for each user and has an advantage over TKEKAS and DKEKAS since the keys are originally dependent but can be converted to independent keys. For all these schemes, key derivation is a single-step process. In Iterative Key Encrypting Key Assignment Scheme (IKEKAS), key generation and updates are relatively easy, but the key generation process is iterative. The article emphasizes that changes in information flow policies must be accounted for in key assignment schemes and highlights the need for research addressing key update and change challenges. Best practices for any key assignment scheme include minimizing private and public storage requirements, employing computationally efficient methods for key derivation and updates, and ensuring $KR_s$ compliance (preventing unauthorized users from deriving keys through collaboration).

A key assignment/access control scheme based on arbitrary access graphs for hierarchical infrastructures was proposed in [18]. This dynamic and efficient scheme utilizes hash functions for deriving child keys from a node's key and restricts the derivation of child class keys to a certain number of linear bit operations. Each class in the scheme is associated with a unique key, ensuring both $KR_s$ and $KI_s$. Each class has a single key associated with it, and the scheme ensures both $KR_s$ (resistance to unauthorized key derivation through collaboration) and $KI_s$ (resistance to key inference). The scheme's security is grounded in the use of pseudo-random functions and a secure symmetric key encryption algorithm. The worst-case time complexity for key derivation is $O(n)$, where $n$ represents the number of nodes in the graph. This is more efficient than earlier schemes, as it avoids computationally expensive operations such as interpolating polynomials and modular exponentiation. As demonstrated in [25], an increased amount of public information results in more edges and classes in the graph, while a greater number of classification levels between classes increases the time cost for key derivation. Similar to the scheme proposed in [18], the scheme presented in [26] is well-suited for dynamic structural changes, including class deletions and insertions.

Various key assignment schemes leveraging elliptic-curve cryptography (ECC) have been proposed to address hierarchical access control problem, including those presented in [27–29]. The scheme presented in [27] is slower than both [19,20] due to the principle that the number of access control policies is related to the number of keys and tamper-resistant machines. However, according to [30], this scheme is not $KR_s$. Similarly, the scheme proposed in [28] is not secure against exterior root-finding attack. The method described in [29] is also not $KR_s$ [31].

The schemes proposed by [32,33] have shown improved performance when compared to their predecessors. [32] introduced a HKA scheme based on secure symmetric encryption, which offers $KI_s$ with more computationally efficient method for key derivation/update when compared to the Atallah et al.'s scheme [18]. Private storage needs are minimal, requiring only one key per class, but an increase in the public storage requirements leads to a larger number of edges and classes in the graph. On the other hand, the scheme proposed in [33] provides efficient computation and storage costs compared to its predecessors.

The schemes proposed in [21] are $KI_s$ and $KR_s$, utilizing symmetric encryption and bilinear maps, respectively, and are provably secure. These schemes feature highly efficient key derivation operations, requiring only a single decryption or pairing evaluation, regardless of the number of levels in the hierarchical structure. Updates to the structure involve changes only to public information, while private information remains unaffected. However, as noted in [25], since these schemes depend on both the number of time periods and classes, the private information required can grow significantly with the number of time periods.

The methods described in [25] rely on forward secure pseudorandom generators (FSPRG) and pseudo-random functions (PRF). Key derivation efficiency and the requirement for private information storage are two factors that are weighed

against each other. Key derivation efficiency is primarily determined by the longest poset depth, while private information needs depend on the poset width. However, key derivation efficiency is generally better compared to other methods. Furthermore, these methods do not require public storage, which is another advantage.

The approach presented in [34] utilizes linear geometry to arrange users in a hierarchical structure, dividing them into distinct security classes to establish varying access privileges. To derive a security class's key, both its own public vector and the parent class's private vector must be employed. The key of the child security class can be directly derived by the parent security class, which eliminates the need for iterative computation and ensures $SKI_s$. The computational cost is manageable, involving vector multiplication and pseudo-random function values. However, this scheme demands greater public storage than others, which is a major disadvantage. Nonetheless, there is a trade-off between storage requirements and computation costs. The approach provides fine-grained control and adaptability to changes/updates in the hierarchy. If modifications occur, the data controller must compute and distribute a new public matrix, which may eventually result in an overall inefficient and intolerable cost. Another drawback is that the matrix must satisfy specific properties to establish a relationship between the hierarchical structure's number of classes and public information, especially for re-keying.

The method outlined in [35] uses a secret sharing algorithm and polynomial interpolation technique. In this approach, the data controller comprises organizational units, and the key is confidential information unique to each unit in the structure. The scheme caters specifically to dependent organizational units [36]. This means that only users with the necessary authorizations from users with higher or equal privileges can access the key using a directed graph's topological sorting, which includes self-loops. The time cost of key derivation, as well as the private and public storage requirements, are acceptable. Additionally, since the organizational unit's secret key does not need to be stored anywhere, the risk of a data breach due to key disclosure is minimized or eliminated.

Recent work by Najafi and Babaie [37] proposes an energy-aware hierarchical key management scheme for IoT, where key server selection and clustering reduce overhead and improve network lifetime. The approach generates shorter keys based on node position and residual energy and, via formal analysis in the ROR model (supported by informal analysis), argues perfect forward secrecy and resistance to impersonation, man-in-the-middle, replay, and key-guessing attacks. However, it assumes reliable position/energy (and neighbor-distance) information for server selection and clustering, and the designated key server remains a critical component whose failure or compromise could disrupt key distribution.

A formal framework incorporating periodic rotation of both secrets and encryption keys into hierarchical key assignment was introduced in recent work [38]. The study proposes an efficient rotation mechanism (KR-HKAS), where the trusted authority updates public information while classes locally evolve their secrets per time period, avoiding setup reinitialization and ongoing redistribution over secure channels. While conceptually strong and backed by provable-security notions, the contribution is primarily theoretical and does not provide implementation-level benchmarks or realistic deployment evidence for system-level overheads. The IHIBE framework integrates IOTA, a lightweight distributed ledger for IoT, with hierarchical identity-based encryption to enable delegated and scalable access control [39]. Its evaluation on AWS and Raspberry Pi 4 confirms scalability across different hierarchy depths and device capabilities. However, the reliance on IOTA interactions introduces high upload latency, limiting the framework's practicality for real-time or large-scale IoT deployments.

A chaotic map-based key management scheme for hierarchical access control in e-medicine systems was introduced in [40]. The approach uses the lightweight and unpredictable properties of chaotic functions to improve efficiency. However, the security evaluation relies mainly on BAN-logic-based reasoning and attack-oriented arguments rather than a standard provable-security model, and it provides limited empirical evidence on scalability for very large hierarchies.

A recent study introduced the Improved Hierarchical Key Management (IHKM) scheme to enhance security and efficiency in hierarchical WSNs [41]. The scheme combines symmetric and asymmetric cryptography to resist various attacks and improve network lifetime. However, its security evaluation is primarily based on an adversarial model and

attack-specific arguments rather than a formal proof framework, and the authors note potential scalability challenges and resource constraints in large networks.

Hierarchical key assignment and access control schemes can be broadly classified into two main categories: direct control and indirect control. Direct schemes, such as those outlined in [25,32,33], require only a single computational step to derive the secret key of a child class. In contrast, indirect schemes,including those presented in [26,28,31,34], derive a child class's secret key by computing allkeys along the hierarchical path to that class. However, as noted in [42–44], indirect schemes are often insecure and susceptible to various attacks. Additionally, these schemes incur significant overhead costs, as highlighted in [31]. This reveals a gap in the current research landscape, indicating the need for a practical and secure key access mechanism. To address this, we are motivated to propose a cloud-independent hierarchical key assignment scheme based on inner product spaces that ensures both security and efficiency.

A comparison of the principal hierarchical key assignment schemes is provided in Table 1. The table outlines the core methodology, advantages, and known limitations of each construction, and highlights (through the final row) how our proposed inner-product–based scheme addresses these issues.

## Preliminaries

### Inner product space

Linear algebra has found numerous applications over the past century [45]. Tools such as matrices, eigenvalues, and linear systems [46] are essential components in computational science, particularly in fields like artificial intelligence and machine learning. Most vector spaces are equipped with a well-defined inner product, which serves as a tool to measure the distance between two vectors within the space. Recently, such spaces have been increasingly employed in the security domain [7]. Once the notion of distance is defined within a space, identifying the closest vectors to specific subspaces becomes a computational task. The algorithm proposed in this work leverages this fundamental concept of distance, expressed through the inner product. A brief overview of the inner product and its properties on a vector space $\Omega$ is provided below. Following this, the procedure for finding the closest vector in a defined subspace is outlined. Since the process involves constructing an orthonormal basis for the subspace, the Gram-Schmidt (GS) orthogonalization process for generating such a basis is briefly explained. Lastly, we describe the final step, which involves computing the orthogonal projection of a vector onto a defined subspace of $\Omega$.

The vector space $\Omega$ equipped with an inner product is referred to as an inner product space. Let $W$ be a subspace of $\Omega$ with a basis $B = \{w_1, w_2, \ldots, w_n\}$. The GS method can be applied to transform $B$ into an orthogonal basis $B' = \{w'_1, w'_2 \ldots, w'_n\}$ for $W$. The orthonormal basis, $B''$, for $W$ is then obtained by normalizing each vector in $B'$, resulting in a set of unit vectors.

**Remark 1.** *In practical applications, we employ the Gram-Schmidt (GS) algorithm with a high-precision setting. We also observe that the Modified Gram-Schmidt (MGS) algorithm effectively mitigates numerical instability. However, implementations in real-valued spaces $\mathbb{R}^m$ may still be susceptible to floating-point rounding errors, particularly in constrained or cross-platform environments. Furthermore, tests conducted with the standard Gram-Schmidt method indicate that increasing precision to 30 decimal digits or more is sufficient to ensure numerical stability.*

### Orthogonal Projection (OP)

In real-time applications, it is often necessary to approximate the value of a function $f(x)$ when this value cannot be computed analytically. For instance, Taylor approximation enables the representation of a continuously differentiable function as a linear combination of polynomials. Similarly, Fourier expansion provides a method to express any periodic function as a linear combination of trigonometric functions. The fundamental concept underlying Fourier expansion is the well-established principle of OP.

**Table 1**. Comparison of principles hierarchical key assignment schemes.

| Schemes | Methodology/Core Technique | Advantages | Limitations/Known Weaknesses | How Proposed Scheme Improves |
|---|---|---|---|---|
| Akl–Taylor [2] | Poset-based hierarchy; modular arithmetic | Simple; foundational model | High public storage; no temporary access; expensive updates; only $KR_s$ | Provides basis subsets; public info independent of hierarchy size |
| MacKinnon et al. [7] | Canonical assignment; optimized public info | Reduced public storage | Still large storage; downward derivation; only $KI_s$ | Subspace-based derivation prevents downward leakage |
| Sandhu [8] | Tree-based symmetric key; one-way function | Fast insertion; simple ops | Expensive derivation for deep trees; limited scalability | Supports arbitrary posets; basis-driven derivation avoids tree-depth cost |
| Harn et al. [9] | Bottom-up derivation; fewer parameters | Lower public storage | Still costly rekeying; limited flexibility | Selective subtree-only rekeying with OP mechanism |
| Chang–Hwang; Liaw et al. [10,11] | Newton method + one-way functions | Conceptually simple | High computational cost; collusion attacks | Resilient to collusion via basis-isolation |
| Liaw–Lei [13] | Up-bottom tree-based heuristic | Low storage; efficient derivation | Limited to trees; cannot handle posets | Handles general posets naturally via subspace nesting |
| Hwang [14] | One-way function + poset support | Supports non-tree structures; $KR_s$ | Storage grows with poset width | Supports arbitrary width; no public info growth |
| Tzeng [17] | Lucas functions; time-bound keys | Time-based access; reduced public info | Computationally expensive; not $KI_s$ under 3-user collusion | Provides time-independence; no collusion leakage |
| Chien et al. [18] | Tamper-resistant hardware; hash functions | High security; $KR_s$ | Requires hardware; vulnerable to 3-user collusion | Software-only; no TTP; collusion resistance via subspace isolation |
| Atallah et al. [16] | Hash chains; access graphs | Dynamic; efficient; $KR_s$, $KI_s$ | Storage increases with graph complexity | Fixed public vectors $(f,g)$ independent of hierarchy |
| ECC-based schemes [25–27] | Elliptic-curve ops | Small keys; strong crypto assumptions | Some not $KR_s$; root-finding attacks | Independent of ECC; no algebraic attack surface |
| Sant–Ferara [31] | Symmetric encryption; tree heuristics | Efficient updates; $KI_s$ | Public storage grows rapidly | No public storage; basis sets only |
| Ragab–Hassen [32] | Optimized symmetric approach | Low computation/storage | Still dependent on graph depth/width | Complexity independent of hierarchy shape |
| Najafi et al. [45] | Energy-aware; cluster-based key-server selection | Low energy use; low delay; long network lifetime | Relies on accurate position/energy data; key-server is a single point of failure | No reliance on node metrics; no single-point-failure; formal projection-based security |
| Cimorelli et al. [46] | PRP-based secret rotation over a plain HKAS | Avoids reinitialization; redistribution | Theoretical; No benchmarks or deployment evidence | Practical overhead analysis; no rotation dependency |
| Purnama et al. [47] | IOTA-integrated hierarchical IBE with delegated key derivation | Delegated; scalable access control | Not practical for real-time or large-scale IoT | No IOTA; low latency; scalable for real-time use |
| **Proposed Scheme** | Inner-product spaces; orthogonal projection; basis isolation | Low storage (only 2 public vectors); efficient OP-based key derivation; scalable for arbitrary posets; collusion-resistant; selective rekeying | None observed for standard threat models; numerical stability depends on implementation choice | Provides $\mathcal{O}(n_i^2)$ key derivation; no public storage growth; strong $SKI_s$; rekeying affects only descendants; suitable for large hierarchies |

Assume that $f$ is a vector in $\Omega$. Then we denote the unique orthogonal projection of $f$ onto $W$ by $\text{Proj}_W f$. Basically, OP addresses the problem of finding the closest vector in the subspace $W$ to a given vector $f$.

A vector $h$ is the nearest vector to $f$ in $W$ if and only if $(f-h)$ is orthogonal to all vectors in $W$. In practice, determining $h$ for a given $f$ is straightforward if an orthonormal basis for $W$ is available. Specifically, the projection vector of $f$ onto $W$

can be expressed as a linear combination of the elements in the orthonormal basis $B''$. Let $h$ represent the closest vector to $f$; it can be computed using the following formula:

$$h = \sum_{i=1}^{n} \lambda_i w_i'' \text{ such that } \lambda_i \text{ in the base field } \mathcal{F} \text{ and } w_i'' \in B''. \tag{1}$$

As $f-h$ must be orthogonal to each one of $\{w_1'', w_2'', \dots, w_n''\}$, then $\langle f - h, w_i'' \rangle = 0$ for $i = 1, \dots, n$ and this implies

$$\lambda_i = \langle f, w_i'' \rangle \text{ for } i = 1, \dots, n. \tag{2}$$

It is important to note that for any orthonormal basis $B''$ of $W$, the OP of $f$ onto $W$ remains the same. Once a basis of $W$ is known, the unique OP can be efficiently computed using the GS orthogonalization process. Fig 3 illustrates the concept of OP in an inner product space, providing a visual representation of how the closest vector $h$ to $f$ is identified within the subspace $W$.

A projection of a vector $f$ onto $W$ is unique. This property ensures that, while each user associated with the subspace $W$ may have a distinct basis, all users can compute the same OP of $f$ onto $W$. This consistency in the computed projection arises from the fundamental nature of OP, which is independent of the specific choice of basis for $W$.

**Remark 2.** *Let $W$ be a subspace of $\Omega$. If we have an orthonormal basis $\beta = \{b_1, \dots, b_n\}$ for $W$, the projection of a vector $f$ onto $W$ can be computed directly as:*

$$Proj_W f = \sum_{i=1}^{n} \langle f, b_i \rangle b_i \tag{3}$$

*Thus, when an orthonormal basis is available, there is no need to perform an orthogonalization process, as the projection formula follows naturally.*

## The proposed scheme

This section presents a comprehensive system model for the proposed hierarchical key assignment scheme, which ensures secure and flexible access control in a structured environment. The model is designed to support a partially ordered set of user groups, where each group is assigned a basis set derived from a higher classification level, adhering to the LRBU policy. The security of the scheme is enforced by leveraging inner product spaces and vector subspaces, allowing efficient key derivation while preventing unauthorized access. The model consists of several key phases: system

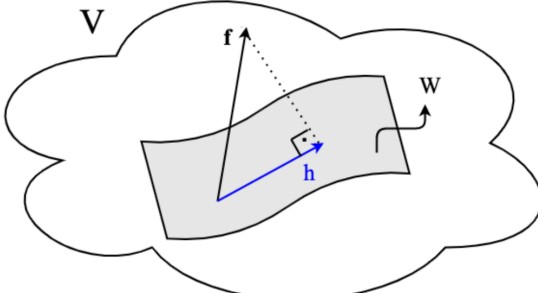

**Fig 3**. **Orthogonal Projection: The closest vector** $h \in W$ **to** $f \in \Omega$ **is the** $Proj_W f$. In other words, $h$ is the nearest vector to $f$ in $W$ if and only if $(f-h)$ is orthogonal to all vectors in $W$.

initialization, hierarchical key assignment, dynamic updates, entropy-aware basis subset selection, key update operations, and key derivation. During initialization, classification levels and basis sets are determined for each group, ensuring hierarchical integrity. The hierarchical key assignment phase distributes group-specific secret keys while maintaining security constraints. The dynamic update phase allows for efficient insertion and deletion of groups without compromising the system's integrity. Additionally, the key update phase enables seamless re-keying during structural changes while preserving forward and backward secrecy. Finally, the key derivation mechanism ensures that authorized users can compute their own keys as well as those of subordinate groups, while preventing unauthorized key inference. Table 3 also includes to quantitatively evaluate the trade-offs in update complexity and trust re-establishment across the scheme. The following sections provide a detailed breakdown of each phase, outlining the mathematical structure and operational workflow of the proposed model.

## Basic rules

Our proposed scheme is based on the following rules.

**Rule 1.** Let the set of user (access control) groups be $G = \{G_1, G_2, \dots G_c\}$. $G_1$ is the most privileged (root) group and $c \geq 1$ is the number of groups in the hierarchical organization.

**Rule 2.** Assume that $(G, \leq)$ is a partially ordered set (poset). For the groups, $G_i$ and $G_j$, $i \leq j$ means that user of $G_i$ can access own secret key $K_{G_i}$ and also $K_{G_j}$ belonging to $G_j$ whose classification level is lower than $G_i$. Note that if $G_i$ is one of the parent groups but $G_i$ and $G_j$ are not in the parent-child relationship like in Fig 1, then $K_{G_j}$ should not be obtained by the members of $G_i$.

**Rule 3.** Let $\Omega$ be an inner product space over a field $\mathcal{F}$ where $\mathcal{F}$ is preferably non-finite field. There is a subspace $W_1$ of $V$ with basis $B_1$ defined for $G_1$. For all other remaining groups in the organization, basis sets $\{B_2, B_3, \dots, B_c\}$ are identified which are all derived from $B_1$. The basis sets $B_i$ for $i = 1, \dots, c$ are designed in comply with the LRBU policy to provide flexibility and efficiency, especially for any dynamic update/change in the structure of the organization.

**Rule 4.** Let assume that $B_1 = \{w_1, w_2, \dots, w_{n_1}\}$ with $W_1 = span\{B_1\}$ is given for poset based access control structure depicted in Fig 1. Note that the vectors $w_1, w_2, \dots, w_{n_1}$ in $B_1$ are linearly independent. The defined basis sets for groups with the same classification (security) level have the same lengths (the number of vectors they contain), which are denoted as follows:

$$|B_1| = n_1.$$
$$|B_2| = |B_3| = n_2.$$
$$|B_4| = |B_5| = |B_6| = |B_7| = |B_8| = |B_9| = n_3.$$

with $n_1 - 1 > n_2$ and $n_2 - 1 > n_3$, namely $n_j - 1 > n_{j+1}$ where we illustrate the key distribution based on the structure given in Fig 1.

For each group, an element in its basis set is selected to be its secret vector. Let the secret vector for $G_1$ be $s_1 \xleftarrow{r} G_1$. Then, the basis sets $B_2, B_3$ for $G_2, G_3$ respectively are obtained as follows:

There are $C(n_1 - 1, n_2)$ distinct subsets of $B_1$ with exactly $n_2$ number of elements and none of these contain the secret vector of $G_1$. Let's assume the set of these subsets are denoted by $B_1^{C(n_1-1,n_2)}$.

$$B_2, B_3 \xleftarrow{r} B_1^{C(n_1-1,n_2)}.$$

This means that $B_2$ and $B_3$ are two randomly chosen subsets from set $B_1^{C(n_1-1,n_2)}$.

Similarly, $B_i$ with $i = 4, \ldots, 9$ are formed as follows:

$$B_4, B_5, B_6 \xleftarrow{r} B_2^{C(n_2-1, n_3)}.$$

$$B_7, B_8, B_9 \xleftarrow{r} B_3^{C(n_2-1, n_3)}.$$

In this set up, every basis set $B_i$ has at least one secret vector $s_i$ that is not shared with its children. Therefore, even if all the children collude and combine their private data, they can not generate their parent's basis set and so they can not derive the parent group's secret key. On the other hand, any parent group $G_i$ has all the required information to derive its children's secret keys.

**Rule 5.** Let's denote $W_i = span\{B_i\}$ with $i = 1, \ldots, 9$. Notice that each of $W_2, \ldots, W_9$ is a subspace of $W_1$. Similarly, $\{W_4, W_5, W_6\}$ and $\{W_7, W_8, W_9\}$ are subspaces of $W_2$ and $W_3$ respectively. The dimensions of the subspaces defined for groups with the same security level are the same, for example; dim $W_2 =$ dim $W_3 = n_2$.

For any $G_i, G_j$;

- If $G_i \leq G_j$, $B_j$ is proper subset of $B_i$ ($B_j \subset B_i$) and $W_j$ is subspace of $W_i$.
- If $G_i \nleq G_j$, $B_j \not\subset B_i$ and $W_j$ is not subspace of $W_i$.

**Rule 6.** Any user of $G_i$ can access $K_{G_i}$ and $K_{G_j}$ (if $G_i \leq G_j$). For example, a user $G_3$ can access the corresponding keys $K_{G_3}$ and also $K_{G_7}, K_{G_8}, K_{G_9}$ but any member of $G_7, G_8, G_9$ which are lower classification level groups, cannot access $K_{G_3}$. They are restricted to accessing only their own secret keys.

The formal definition of a hierarchical key assignment scheme is provided below [8,18,25,34].

A hierarchical key assignment scheme for $\Gamma$ (a set of access graphs) is defined as a pair of two polynomial-time algorithms, **Set** and **Derive**, described as follows:

- The information generation algorithm **Set($1^\rho$, $\mathfrak{G}$)** is probabilistic and takes as input $1^\rho$ (security parameter) and an access graph $\mathfrak{G} = (G, E)$ in $\Gamma$ and produces: (1) for each group $G_i \in G$; $B_i \subset B_1$ and $K_{G_i} \in \{0, 1\}^\rho$, (2) public information (vectors) ($f$,$g$).
  Let the output of **Set($1^\rho$, $\mathfrak{G}$)** be symbolized as ($B$,$K$,$f$,$g$).
- The key derivation algorithm **Derive($1^\rho$, $\mathfrak{G}$, $B_i$, $f$, $g$)** is deterministic and takes a security parameter $1^\rho$, a graph $\mathfrak{G}$, private set $B_i$, and public informations $f$,$g$. The algorithm's return values are as follows.

$$\text{Derive}(1^n, \mathfrak{G}, B_i, f, g) = K_{G_i}. \tag{4}$$

and also $K_{G_j}$ iff $G_i \leq G_j (B_j \subset B_i)$.

## System-ready phase

The system preparation and distribution processes are outlined in Algorithm 1 and illustrated in Fig 4, which illustrates the sequential steps involved in the system-ready phase for secure hierarchical key management. The process begins with the definition of classification levels, groups, and the hierarchical structure according to the LBRU policy. A dimension policy is then established, ensuring that each classification level is assigned unique basis sets to maintain security and hierarchical integrity. Following this, a basis is constructed for each group across classification levels. To enhance security, random public vectors ($f$,$g$) and user-specific random constants are generated. Using these, a unique basis is constructed for each user by multiplying the group basis with the assigned random constants. Once these steps are completed, the system becomes fully operational and ready for secure hierarchical key management.

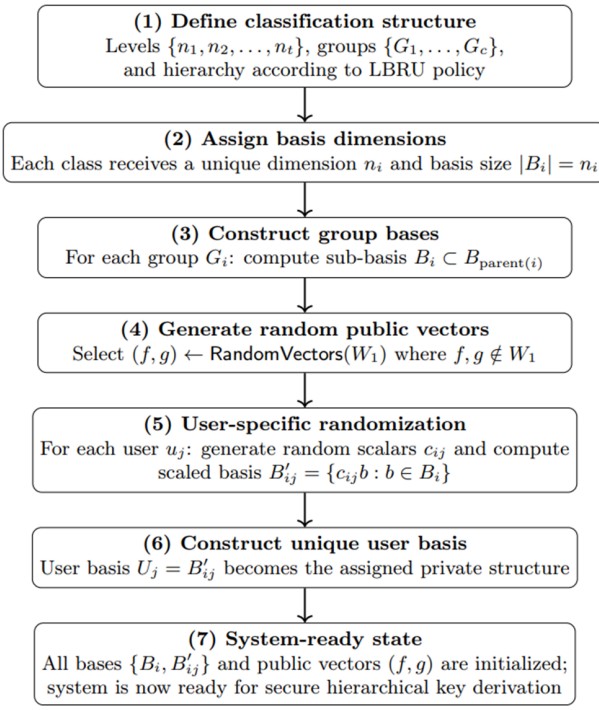

**Fig 4**. Steps of the system-ready phase.

**System preparation phase.** Algorithm 1 initializes the hierarchical key assignment by defining groups $G = \{G_1, G_2, \cdots, G_c\}$, selecting an inner product space $\Omega$ over a field $\mathcal{F}$, and constructing basis sets $B_i$. The root group $G_1$ is assigned $B_1$, while each child $G_i$ receives $B_i \subset B_{\text{parent}(i)}$ to enforce hierarchical security. Public vectors $f, g \in \Omega$ enable secure key derivation, ensuring that privileged groups can compute subordinate keys while preventing unauthorized access.

The data controller:

**Step 1.** Determine all groups $G = \{G_1, G_2, \cdots, G_c\}$ for the organization and their positions in the organizational structure based on Rule 1 and Rule 2 as illustrated in Figs 1 and 2.

**Step 2.** Determine a vector space $\Omega$ that may have infinite dimension over a field $\mathcal{F}$, and define an inner product on $\Omega$. It is worth noting that, although the polynomial space over the real numbers could be considered a suitable candidate, in practical implementations, we consider the finite extension $\mathbb{R}^m$ of $\mathbb{R}$.

**Step 3.** Identify basis set $B_1 = \{w_1, w_2, \ldots, w_{n_1}\}$ with $W_1 = \text{span}\{B_1\}$. Determine all basis sets $B_i$ derived from $B_1$ for $i = 2, \ldots, c$ according to Rule 3,4,5.

The selected $\Omega, (B_1, W_1)$ structure allows the data controller to adjust the number of $(B_i, W_i)$ as the number of groups, $c$, in the structure changes. The parameters $n_2, n_3, \ldots, n_t$, used in Algorithm 1, define the dimensions of the hierarchical levels in the key assignment structure.

**Alt Step 3.** The above rules and key distribution scheme are designed for general use. As mentioned earlier, the orthogonalization process, particularly the application of the Gram-Schmidt method, introduces computational errors. To mitigate this, we have implemented our scheme with a precision of up to **30 decimal digits**, which provides sufficient stability in our case.

However, in scenarios where **30-digit precision** is unsuitable or where users wish to avoid instability issues, we introduce a modified version of **Step 3**, referred to as **Alternative Step 3**:

- Identify an orthonormal basis set $B'_1 = \{w'_1, w'_2, \ldots, w'_n\}$ such that $W_1 = \mathrm{Span}(B'_1)$.
- Following rules **3, 4, and 5**, determine all basis sets $B'_i$ based on $B'_1$.
- It is important to note that the basis sets for the other groups remain **orthonormal**, as they are derived from the initial orthonormal set $B'_1$.

**Distribution phase.** In this phase, the data controller distributes the public and private information to the relevant groups $G_i$, enabling each member $u_{ij}$ to derive the corresponding key or keys in accordance with Rule 6.

**Step 1.** Select two random vectors $f, g \in V$ such that $f, g \notin W_1$ which also implies $f, g \notin W_i$ for $i = 2, \cdots, c$ and makes them known to the public.

**Step 2.** For each user $u_{ij}$ in the group $G_i$ select a random constant $c_{ij} \in \mathcal{F}$ and set $B_{ij} = \{c_{ij}w_1, \ldots, c_{ij}w_n\}$.

**Alt Step 2.** This alternative step is designed specifically to avoid high-precision numerical operations. If, during the system initialization phase, an orthonormal basis is constructed for each space, then in this alternative approach, the user $u_{ij}$ receives an orthogonal basis set:

$$B'_{ij} = \{\, c_{ij}w'_1,\ c_{ij}w'_2,\ \ldots,\ c_{ij}w'_n \,\}. \tag{5}$$

If **Alternative Step 2** is implemented, all users can disregard the orthogonalization process.

**Step 3.** Distributes the basis set $B_{ij}$ derived from the basis set $B_i$ to each member of the group $G_i$ based on LRBU policy.

Fig 5 provides an overall workflow diagram of the proposed hierarchical key assignment scheme, combining the System-Ready Phase (steps 1a–1b), the Key Derivation Phase, and the Dynamic Update/Rekeying Phase. This figure visually summarizes the processes formalized in Algorithm 1 and Algorithm 2, offering a complete end-to-end representation of the system.

**Storage and bandwidth benchmarks during key distribution.** We provide concrete benchmarks for both the memory usage and the communication bandwidth required during the key distribution phase, which are summarized in Table 2.

During this phase, each user $u_{ij}$ receives a scaled basis

$$B'_{ij} = \{c_{ij}w_1, \ldots, c_{ij}w_{n_i}\}. \tag{6}$$

where each vector contains $m$ coordinates of $\lambda$ bits. Thus, the exact memory footprint of the key material assigned to each user is

$$\mathrm{Mem}_{\mathrm{user}} = n_i \cdot m \cdot \lambda \text{ bits.} \tag{7}$$

For the deployment parameters used in our evaluation ($m = 32$, $n_i = 10$, $\lambda = 32$):

$$\mathrm{Mem}_{\mathrm{user}} = 10 \times 32 \times 32 = 10{,}240 \text{ bits} \approx 1.25 \text{ KB.} \tag{8}$$

This compact size demonstrates that the scheme is suitable for resource-constrained devices and low-bandwidth environments.

**Bandwidth requirements.** Since the distribution phase consists of transmitting the scaled basis $B'_{ij}$ to each user, the communication bandwidth equals the size of the transmitted key material:

$$\mathrm{BW}_{\mathrm{dist}} = n_i \cdot m \cdot \lambda \text{ bits.} \tag{9}$$

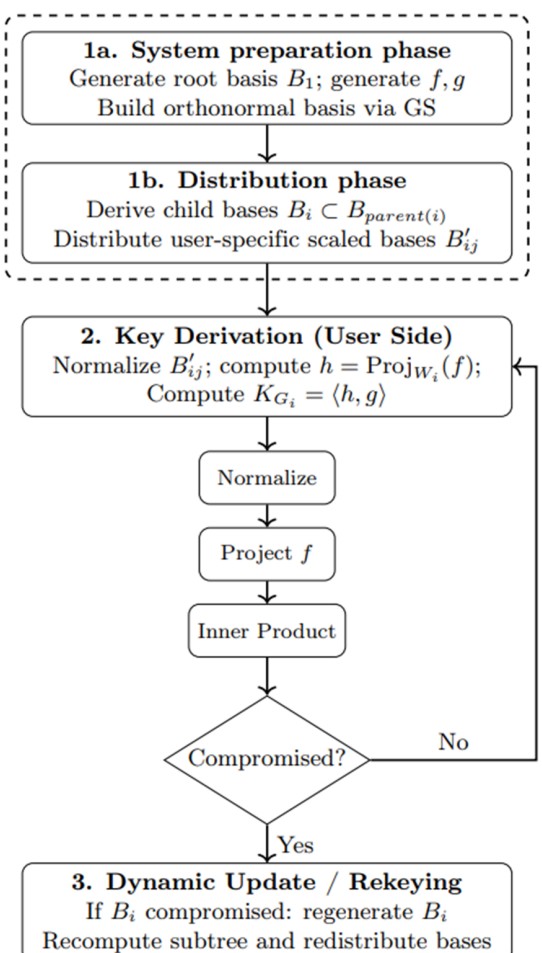

**1. System-Ready Phase**

**1a. System preparation phase**
Generate root basis $B_1$; generate $f, g$
Build orthonormal basis via GS

**1b. Distribution phase**
Derive child bases $B_i \subset B_{parent(i)}$
Distribute user-specific scaled bases $B'_{ij}$

**2. Key Derivation (User Side)**
Normalize $B'_{ij}$; compute $h = \mathrm{Proj}_{W_i}(f)$;
Compute $K_{G_i} = \langle h, g \rangle$

Normalize

Project $f$

Inner Product

Compromised? — No

Yes

**3. Dynamic Update / Rekeying**
If $B_i$ compromised: regenerate $B_i$
Recompute subtree and redistribute bases

**Fig 5**. **Workflow diagram for the proposed hierarchical key assignment scheme.** Steps 1a and 1b are grouped under the System-Ready Phase, followed by key derivation and dynamic rekeying with compromise detection.

**Table 2**. **Memory and bandwidth benchmarks during key distribution.**

| Metric | Value |
|---|---|
| Memory per user (key material) | $n_i m \lambda = 10{,}240$ bits $\approx 1.25$ KB |
| Bandwidth per user (initial distribution) | $10{,}240$ bits $\approx 1.25$ KB |
| Selective rekeying bandwidth ($m'$ affected users) | $m' \cdot n_i \cdot m \cdot \lambda$ bits |
| Public vectors broadcast ($f, g$) | $2m\lambda = 2{,}048$ bits |

For the same parameters:

$$\mathrm{BW}_{\mathrm{dist}} = 10{,}240 \text{ bits} \approx 1.25 \text{ KB per user.} \tag{10}$$

**Selective rekeying.** If a basis $B_i$ is compromised, only the descendants of that group (denoted $m'$ users) must be updated. The bandwidth required for rekeying is therefore:

$$\mathrm{BW}_{\mathrm{rekey}} = m' \cdot n_i \cdot m \cdot \lambda \text{ bits.} \tag{11}$$

This linear dependency on the number of affected users ensures that rekeying traffic remains minimal, even in large hierarchies.

## Key derivation phase

To derive the corresponding secret key, a user $u_{ij}$ in the hierarchical structure performs the following tasks (see Algorithm 2):

**Step 1.** Apply GS orthogonalization operation to its basis $B_{ij}$, namely computes corresponding orthonormal basis $B''_{ij} = \{w''_1, w''_2, \ldots, w''_n\}$.

**Alt Step 1.** If users have already received an orthonormal basis as their secret key, they do not need to go through the Gram-Schmidt process. Since the received set is already orthogonal, users only need to normalize each vector. During the key generation step, each vector is multiplied by a constant $c_{ij}$. The normalization step simply removes this constant to obtain the final orthonormal basis.

**Step 2.** Runs OP to compute $h$ (See Fig 3), which is the closest vector to $f$ in $W_i$.

$$h = (\text{Proj}_{W_i} f) = \sum_{i=1}^{n} \langle f, w''_i \rangle w''_i \qquad (12)$$

The secret key $K_{G_i} = \langle h, g \rangle$.

Notice that if a member of $G_i$ needs to construct another key, such as $K_{G_j}$, they must possess a basis for the corresponding subspace $W_j$. Therefore, if a child wants to access its parent's secret key, it must also have a basis for its parent's subspace, which cannot be derived or guessed from the child's own basis. For example, if $W_1$ is a $n_1$-dimensional subspace of $\mathbb{R}^m$ and $W_2$ is a subspace of $W_1$ with $n_2$-dimension ($n_1 - 1 > n_2$), it is infeasible to reconstruct $W_1$ from $W_2$. In fact, there are uncountably many subspaces of $\mathbb{R}^m$ with $n_1$-dimensions, and knowing only $n_2$ basis elements does not allow the derivation of the remaining elements. For further details, see [7].

## Dynamic update phase

Our proposed hierarchical key assignment/access control scheme should enable the adaptation of changes in the access control structure topology over time. Specifically, it allows for the insertion/deletion of groups at any classification level without the requirement to construct new public information ($f,g$) utilized by any $G_i$. Additionally, the computational cost and need for private information during the insertion/deletion process are negligible.

**Insertion.** With our proposed scheme, one or more groups can be added to the hierarchical structure. A group addition to an existing classification level incurs the same computational cost as selecting a random subset from set $B_i^{C(n_j-1, n_{j+1})}$. Each added group is set as a subgroup of a group with a higher classification level. In addition, one or more groups can be added as children of a group with the lowest classification level. In this case, new classification (security) levels are created for each added group.

Let us continue with adding new groups to different security levels in the access scheme we illustrated earlier in Fig 1 (see Fig 6). In the access scheme shown in Fig 1, there are 3 security levels, which are top secret, secret, and confidential.

In order to add $G_{u2}$ to the confidential classification level (as a subgroup of $G_3$) ;
We know that there are

$$\left( \frac{(n_2 - 1)!}{n_3!(n_2 - n_3 - 1)} - 3 \right).$$

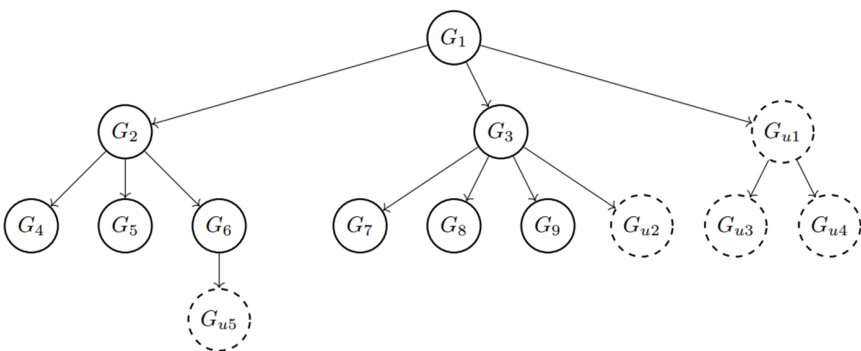

**Fig 6. Insertion of $G_{u1}, G_{u2}, G_{u3}, G_{u4}, G_{u5}$ into the organization structure.**

distinct subsets that can be used as the basis set of a group $G_{u2}$. The $G_{u2}$ basis set is established as one of these randomly selected subsets as follows.

$$B_{u2} \xleftarrow{r} \left[ B_3^{C(n_2-1,n_3)} - \{B_7, B_8, B_9\} \right].$$

Then Algorithm 1 is executed by the data owner starting from step 12.

Similarly, the groups $G_{u1}, G_{u3}, G_{u4}$ are recursively included in the access scheme. This means that group $G_{u1}$ is added first, then groups $G_{u3}$ and $G_{u4}$. The basis set for each group is obtained as follows;

$$B_{u1} \xleftarrow{r} \left[ B_1^{C(n_1-1,n_2)} - \{B_2, B_3\} \right].$$
$$B_{u3}, B_{u4} \xleftarrow{r} B_{u1}^{C(n_2-1,n_3)}.$$

Again, Algorithm 1 is executed by the data owner from step 12. Finally, $G_{u5}$ is added to the hierarchy. The basis set for the $G_{u5}$ is obtained as follows;

$$B_{u5} \xleftarrow{r} B_6^{C(n_3-1,n_4)}.$$

The important thing here is that a new classification level can be named as restricted and it is created for group $G_{u5}$.

We would like to point out that the structure of the organization can also be preset to create and insert a new classification level. The placeholder values of $n_i$ can be used for this purpose.

**Deletion.** The data owner can delete one or more groups from its organization. If there is no longer a group in a classification level, the classification level can also be deleted. If a parent $G_i$ is deleted from the structure, all children of the deleted parent are recursively linked to a higher parent group and the classification level of each child is updated. New basis sets are selected recursively for all groups whose security level has changed (moved to a higher security level). Old basis sets used by these groups are overridden and not assigned to the newly added groups. However, if the deleted group is at the bottom of the hierarchical structure, there will be no change in the linkage. Basis sets of deleted groups are not used for groups that are newly added to the hierarchy.

**Key Update.** Each group $G_i$ possesses a unique secret key $K_{G_i}$. If a key is compromised or requires replacement, the data owner re-executes Algorithm 1, assigning new public parameters $(f, g)$. Conversely, if a group's basis is compromised, the data owner initiates Algorithm 1 from Step 8 for $G_i$ and its descendants. This process updates the relevant basis sets, ensuring continued consistency within the hierarchy.

A summary of the computational and trust-related costs for dynamic operations is provided in Table 3. Unlike schemes that rely on external trusted third parties (TTPs), our design assumes that only the system controller is trusted to perform dynamic updates and key management operations. No additional trusted infrastructure is required.

## Experimental results

The proposed hierarchical key access scheme was implemented on a system running Macintosh OS Ventura with a 3.2 GHz 6-Core Intel Core i7 processor and 16 GB 2667 MHz DDR4 memory. The implementation utilized SageMath and Jupyter Notebook for computational tasks and experimentation.

Algorithm 1 constructs hierarchical basis sets, while Algorithm 2 derives hierarchical keys using inner product operations. During the system initialization phase, a random subspace of the universal space $\Omega$ is selected. If the system aims to mitigate instability issues in computation, an alternative method for basis generation is employed. Specifically, in this phase, the owner constructs a random orthonormal set.

This can be achieved in several ways. One approach is to select a random basis and apply high-precision arithmetic along with the modified Gram-Schmidt orthogonalization process. However, a more straightforward method involves generating a symmetric matrix $A$ and randomly selecting vectors from distinct eigenspaces. Since the eigenspaces of a symmetric matrix are inherently orthogonal, the choice of $A$ can significantly reduce instability during the orthogonalization process. Once an orthogonal set is determined, constructing an orthonormal set from it becomes trivial.

This alternative approach also reduces the computational burden for each user, as they receive an already orthogonal set. Consequently, during the key derivation phase, users can bypass the Gram-Schmidt orthogonalization process entirely.

The key derivation process for each group $G_i$ is based on Algorithm 3 to perform GS orthogonalization, ensuring numerical stability. To enhance stability during the orthonormalization step, the GS algorithm was employed with precision of 30 decimal digits instead of the classical formulation. This choice significantly improves numerical accuracy, especially when operating in high-dimensional real vector spaces used in the simulation. To improve scalability and computational efficiency, $\Omega$ is defined as an inner product space over a field $\mathcal{F}$. While infinite dimensional spaces like polynomial spaces are useful in theory in practical implementations we use the well-known real vector space $\mathbb{R}^m$ with $m \leq 100$. The vectors $f$ and $g$ should be chosen so that they do not reside in the initial subspace $W_1$. Since the dimension of $W_1$ is selected to be less than $m - 1$, the probability of randomly selecting two vectors that lie within $W_1$ is negligible. However, if the chosen vectors happen to lie in $W_1$, users without the basis for $W_1$ have no method to verify whether the vectors truly belong to the subspace.

**Table 3**. Cost and trust summary for dynamic updates.

| Operation | Cost Description |
|---|---|
| Group Insertion | Subset selection from parent basis: $\mathcal{O}\left(\binom{n_{(j-1)}-1}{n_j}\right)$. |
| Group Deletion | Local re-basing for descendants only; unaffected subtrees remain valid. |
| Rekeying Overhead | Algorithm 1 restarted from Step 8 for the affected subtree only. |
| Public Info Update | Vectors $f$, $g$ changed. |
| Trust Assumption | Only the system controller is required; no external TTP needed. |

**Algorithm 1.**

**Input:**
1  $\Omega$ finite dimension over an infinite field $\mathcal{F}$ and $B_1$ ($|B_1| = n_1$)
2  $\{n_2, n_3, \ldots, n_t\}$, $t$ is the number of classification level
3  $G = \{G_1, G_2, \cdots, G_c\}$, $c$ is the number of groups
4  $\mathbf{u_{ij}}, i = 1, 2, \ldots, c, j = 1, 2, \ldots, k$ , where $k$ is the number of users in $i^{th}$ group
   **System-ready:**
5  $W_1 \leftarrow span\{B_1\}$,  $W_1 \subset \Omega$
6  $f, g \overset{r}{\leftarrow} selectRandomVectors(\Omega)$, where $f, g \notin W_1$
7  **for**  $i = 2$ to $c$  **do**
8    $\quad B_i \overset{r}{\leftarrow} B_{parent(i)}^{C(n_{(j-1)}-1, n_j)}$ , $j$ is the classification level index $G_i$ is in and $B_i \subset B_{parent(i)}$
9    $\quad W_i \leftarrow span\{B_i\}$ is subspace of $W_{parent(i)}$
10   $\quad$ **for**  $j = 1$ to $k$  **do**
11     $\quad\quad c_{ij} \overset{r}{\leftarrow} \mathcal{F}$
12     $\quad\quad B_{ij} \leftarrow c_{ij}B_i$
13     $\quad\quad u_{ij} \leftarrow B_{ij}$

If users might encounter difficulties managing numerical instability while constructing an orthonormal basis, the provided alternative version of Algorithm 1 can be implemented to distribute an orthogonal basis to each user at the beginning.

**Algorithm 1.    Alternative Algorithm 1.**

**Input:**
1  Follow the initial four steps of Algorithm 1.
   **System-ready:**
2  $W_1 \leftarrow span\{B_1'\}$,  $W_1 \subset \Omega$
   - Decide method for orthonormal basis construction:
     - If high-precision arithmetic is preferred:
       - Select a random basis $B = \{v_1, v_2, \ldots, v_n\}$.
       - Apply **Modified Gram-Schmidt** to construct $B'$.
     - Otherwise:
       - Generate a symmetric matrix $A$ in $\Omega$.
       - Compute eigenvectors of $A$.
       - Select $n$ eigenvectors corresponding to distinct eigenspaces to form $B'$.

3  Follow steps 6 through 13 of Algorithm 1.

## Performance analysis

This section analyzes the performance of the proposed scheme by evaluating key metrics, including public and private storage requirements, system readiness, key derivation, and update overheads, and the impact of changes in the hierarchical structure, the compactness of key sizes, the lightweight representation of secrets, the applicability of parameter configurations for practical deployments across varying security levels, and the cost-effectiveness of selective key updates.

**Storage cost for public and private information.**  Random vectors $f, g$ are the public information for the entire system to derive the corresponding secret keys. All basis sets used for the users are derived from $B_1$. Since $|B_1| = n_1$, the

**Algorithm 2.  Key derivation algorithm.**

 **Input:**
1 $B_{ij}$, `f`, `g`
 **Key Derivation:**
2 **If** $B_{ij}$ is not orthogonal
3 $B'_{ij} \leftarrow$ `Modified Gram-Schmidt`$(B_{ij})$
4 $B''_{ij} \leftarrow$ `normalize` $B'_{ij}$
5 **Else**
6 $B''_{ij} \leftarrow$ `normalize` $B_{ij}$
7 $W_{ij} \leftarrow$ `span`$\{B''_{ij}\}$
8 $h \leftarrow$ `Proj`$_{W_{ij}}(f)$
9 $K_{G_i} \leftarrow \langle h, g \rangle$

total number of vectors used by the system is bounded by $n_1$. The basis set $B_1$ is kept secret. Each member $u_{ij}$ of the $G_i$ needs its own basis $B_{ij}$ to derive the secret key $K_{G_i}$. Suppose there are $c$ groups and the maximum number of members in a group is $k$, so the number of basis is at most $ck + 1$.

**System-ready and key derivation time-cost.** The secret key $K_{G_i}$ for a group $G_i$ is derived using the private information $B_{ij}$ and the public information $(f, g)$. The key derivation process may involve the Gram-Schmidt orthogonalization operation unless Alternative Algorithm 1 is implemented. Once a user constructs an orthonormal basis for their subspace, the projection of the public vector $f$ onto the subspace can be efficiently computed using basic inner product operations.

The computational complexity of this process depends on the size of the basis and the inner product defined in the universal space $V$. To evaluate the complexity, we fix the group $G_i$ and its corresponding subspace $W_i$. Let $n_i$ denote the dimension of $W_i$, i.e.

$$n_i = \dim W_i. \tag{13}$$

The complexity of the key derivation procedure (as outlined in Algorithm 2) is analyzed across three primary operations:

- **Number of Inner Products** $I$: Computation of inner products for the basis vectors and the public vector.
- **Multiplication of Constants** $M$: Scalar multiplications required for orthogonalization and projection steps.
- **Division of Constants** $D$: Scalar divisions performed during normalization.

**Total number of inner products $I$**

1. If Alternative Algorithm 1 is used: $2n_i + 1$.
2. If Algorithm 1 is used: $\frac{n_i(n_i-1)}{2} + n_i + 1$.
3. If Alternative Algorithm 1 is not used, the Gram-Schmidt procedure requires:

$$\sum_{j=1}^{n_i}(n_i - j) = \frac{n_i(n_i - 1)}{2}. \tag{14}$$

 inner products.
4. The projection operation (see Eq 1) requires computing the coefficients in Eq 2, contributing an additional $n_i$ inner products for the $n_i$-dimensional vector space.
5. Finally, the last statement in Algorithm 2 necessitates one extra inner product.

**The number of *M* is bounded by** $(n_i + 1)^2$

1. If the Gram-Schmidt process is required, the number of multiplications is:

$$(n_i + n_i(n_i - 1)).$$

2. Algorithm 2 requires a total of $(2n_i + 1)$ scalar multiplications.

**The number of *D*:** $n_i$

1. A total of $n_i$ scalar divisions are required to normalize the $n_i$ basis vectors.

In summary, the complexity of deriving the secret key $K_{G_i}$ for a user is at most $\mathcal{O}(n_i^2)$, with an upper bound given by:

$$\left(\frac{n_i(n_i - 1)}{2} + n_i + 1\right) + (n_i + 1)^2 + n_i.$$

We should note that the heuristic complexity is $\mathcal{O}(n_i)$ when the distribution of the orthonormal basis is chosen in the key generation step.

**Key size and lightweight representation.** The lightweight nature of the scheme is reflected in its efficient key generation. The group secret key $K_{G_i}$ is derived as the inner product of two vectors, one of which is obtained by OP onto the user's associated subspace. As the derivation avoids computationally expensive cryptographic primitives (e.g., computing in elliptic curve groups or pairing-based operations), the runtime and storage overhead remain minimal.

The storage requirement for a group key is given by $\approx (k \times m \times d)$ bits, where $m$ is the dimension of the universal space (i.e., the number of components), $k$ is the bit-length of each component (e.g., 16 bits) and $d$ is the dimension of the corresponding subspace for any group $G'$. For typical values such as $m = 16$, $k = 32$ and $d = 6$ the secret basis set occupies only 3072 bits (384 B).

For $m = 64$, $k = 64$ and $d = 5$, our scheme has a basis of size 20480 bits, which is smaller than Dilithium-5's 38,912 bits. Compared to other schemes like RSA-2048 and ECC, our scheme offers acceptable public and private key sizes, while ensuring efficient post-quantum resistance, making it suitable for constrained environments.

We implemented Algorithm 1 and Algorithm 2, using the real vector space $\mathbb{R}^m$ over the set of real numbers to evaluate real-time computational costs. The inner product for this implementation was chosen to be the dot product, ensuring simplicity and efficiency in the computations.

Experimental results are presented in the following graphs Figs 7, 8, 9, and 10. The binary access structure (BAS) is a type of hierarchical structure in which each node has exactly two child nodes. The ternary access structure (TAS) is a hierarchical structure where each node has exactly three child nodes. The $c$ denotes the number of groups. The reduced number of vectors in a basis results in lower key-derivation costs, which is a crucial factor.

For larger hierarchical structures, such as $c > 121$ with more than 5 classification levels, storage and computational costs naturally increase but remain within practical applicability limits. As the number of groups increases, the dimension of the space also grows, introducing some additional complexity in the key derivation process. However, as shown in Figs 7, 8, 9, and 10, the proposed scheme efficiently manages this growth. While higher-dimensional spaces require more computation, the $O(d^2)$ complexity ensures that the scheme remains scalable and feasible even in deep hierarchies. These results confirm that the proposed scheme can support large-scale access control systems while maintaining efficiency.

**Parameterization guidelines for practical deployment.** To provide practical guidelines for deployment, we recommend the following parameter values for achieving a desired flexibility for dynamic update:

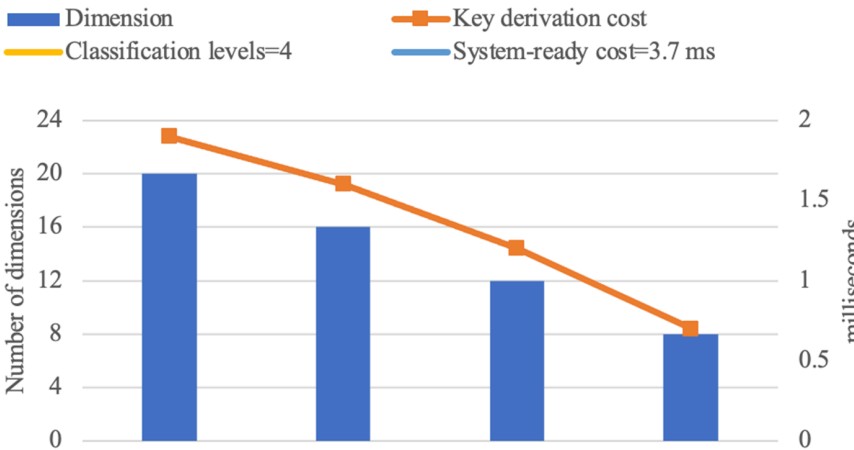

**Fig 7**. **BAS with 4 Classification Levels ($|G|$=$c$=15).**

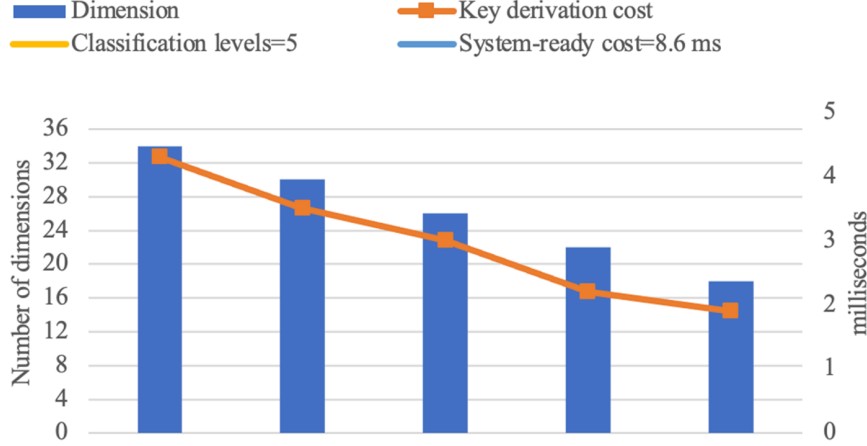

**Fig 8**. **BAS with 5 Classification Levels ($|G|$=$c$=31).**

- $m = 100$: The dimension of the real vector space $\mathbb{R}^m$,
- $n_1 = 64$: Number of basis vectors (dimension) at the top classification level,
- $n_2 = 32$, $n_3 = 16$: Number of basis vectors (dimension) at lower levels,
- $k = 60$: Bit-length per component (e.g., 60-bit fixed-point representation).

Since the inner product operations scale linearly with $m$ and the total key derivation complexity is $\mathcal{O}(m^2)$, the scheme remains efficient and scalable even in large-scale deployments with 100+ user groups.

**Key update.** Each group $G_i$ is assigned a unique secret key $K_{G_i}$. If a key is compromised or needs to be refreshed, the data controller re-executes Algorithm 1 and change the public parameters for $G_i$ and its descendant groups. This targeted rekeying does not affect the global system configuration or public information vectors $(f, g)$, ensuring consistency and minimizing disruption.

In case a basis set is compromised for a group, the update process involves regenerating the basis set for the affected groups and securely redistributing it. Since only a small number of vectors are transmitted per user, the communication overhead scales linearly with the number of updated users and remains lightweight even in large-scale deployments.

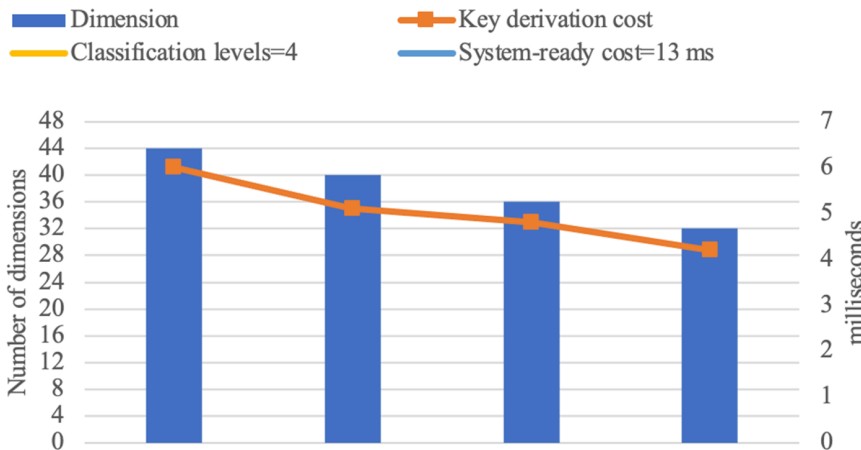

**Fig 9**. TAS with 4 Classification Levels (|$G$|=$c$=40).

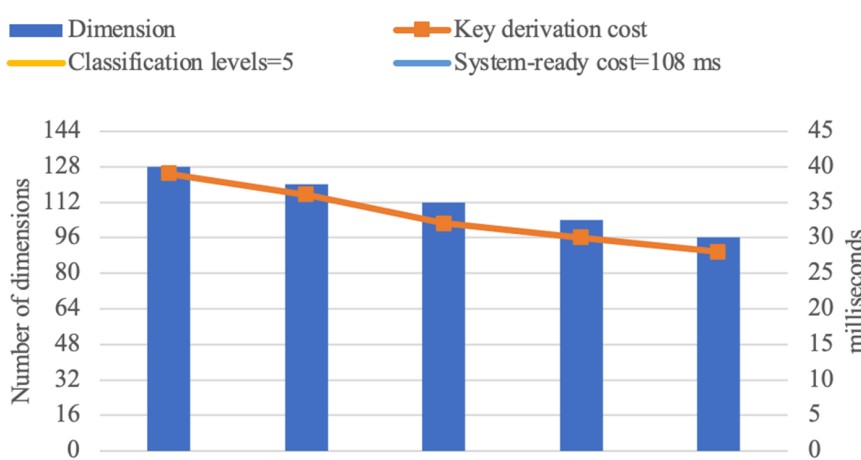

**Fig 10**. TAS with 5 Classification Levels (|$G$|=$c$=121).

From a performance standpoint, this selective rekeying avoids full system reinitialization, reduces computation to localized orthonormal basis generation (Algorithm 3), and eliminates the need for re-deriving unaffected keys. Consequently, the scheme ensures fast recovery from key compromise events with negligible computational and communication cost, preserving overall system efficiency and scalability.

**Communication cost of dynamic updates.** To estimate the communication cost of dynamic updates, we assume that each key or basis set element is represented as a fixed-length bit string of length $\lambda$ (e.g., 256-bit AES keys or group elements). When a rekeying operation is required, the system executes **Algorithm 1 from Step 8** on the affected group and its ancestors. During recomputation, the number of forwarded messages is proportional to the number of affected users in the subtree update. The total communication cost per update is given by:

$$\mathcal{O}(m' \cdot \lambda) \text{ bits}$$

where $m'$ represents the number of affected users. For instance, scenarios such as batched insertions or key compromises within a depth-$d$ subtree may generate multiple update messages per user, depending on the inheritance structure and classification level. However, in practical cases with moderate hierarchy depth ($d<10$) and reasonable key sizes ($\lambda < 512$ bits), the communication overhead remains manageable.

**Trust model extensions.** The current scheme follows a **centralized trust model**, where a system controller manages dynamic updates, key generation, and basis management. While efficient, this model may not be suitable for federated or decentralized systems.

To enhance resilience, the scheme can incorporate **distributed basis generation** using **threshold cryptography** (e.g., Shamir or Pedersen secret sharing). In this approach, a minimum of $t$ out of $n$ partially trusted authorities collaboratively generate or update basis sets and corresponding secrets.

By decentralizing key management, this method:

- Reduces vulnerability to single points of compromise.
- Enhances system security and improves fault tolerance.
- Introduces additional coordination and setup overhead.

Careful consideration is required to balance security improvements with the complexity introduced by distributed key management.

### Security analysis

This section analyzes the security of the proposed scheme, focusing on key challenges such as collusion resistance, key indistinguishability, and privilege creep prevention. The scheme must prevent adversaries from deriving unauthorized keys, even through collaboration or structural inference. Additionally, it ensures that revoked users or downgraded groups lose access after updates. We formally evaluate its resilience against these threats. To support this evaluation, we introduce formal security models such as key recovery security ($KR_s$) and strong key indistinguishability ($SKI_s$), backed by game-based experiments. Furthermore, we analyze the trust and rekeying mechanisms, provide benchmark-based parameter guidelines aligned with NIST levels, and demonstrate the quantum resistance of the scheme .

**Formal resistance to collusion attacks (key recovery security - $KR_s$).** A fundamental security requirement in hierarchical key assignment schemes is collusion resistance, also referred to as $KR_s$ in the literature. In a well-structured hierarchy, it must be computationally infeasible for users in child groups or same-level groups to reconstruct the secret key of their parent or peer groups.

Each group $G_i$ is assigned a unique basis $B_i$ in a vector space $V$, ensuring hierarchical security under the LRBU policy. As established in Basic rules section, the hierarchical structure ensures that lower-level groups lack the necessary basis vectors to reconstruct the secret key of a higher-level group. We consider an adversary $u$ that possesses all public information as well as private information of its chosen group members. In other words, adversary $u$ has the combined capabilities of both a static and an adaptive adversary [47]. The objective of adversary $u$ is to obtain the access key for a higher-level class. For simplicity, we assume that $u$ has private information for all child groups of a given group $G$. The goal is to recover the secret $K_G$ of group $G$. Formally, we analyze the infeasibility of achieving this objective. We first present the formal definition of key recovery security against a static or adaptive adversary, as commonly found in the literature.

**Definition 1.** *Let $\Gamma$ be a family of graphs corresponding to partially ordered hierarchies, and let $\mathfrak{G} = (V, E)$ be a graph in $\Gamma$. Let KeyDistribution be a hierarchical key assignment scheme for $\Gamma$, and let $g$ be an adversary targeting a class G. Consider the following experiment, where $\rho$ denotes the security parameter:*

$\quad$ ***Experiment*** $\textbf{\textit{Exp}}_g^{KeyRecovery}(1^\rho, \mathfrak{G})$

$\quad\quad$ $(B, K_G, f, g) \leftarrow Set(1^\rho, \mathfrak{G})$

$corr_g \leftarrow Corrupt_g(B)$

$K_G \leftarrow Keys_G(K)$

$k_g \leftarrow g(1^\rho, \mathfrak{G}, f, g, corr_g)$

$return\ k_g$

The advantage of the attacker $g$ is defined as:

$$\text{AdvKeyRecovery}_g^{\text{KeyRec}}(1^\rho, \mathfrak{G}) = \Pr[\ k_g = K_G\ ]. \qquad (15)$$

The scheme is considered secure if, for every graph $\mathfrak{G} = (V, E)$ in $\Gamma$ and every group $G \in V$, the function

$$\text{AdvKeyRecovery}_g^{\text{KeyRec}}(1^\rho, \mathfrak{G})$$

is negligible for any adversary $g$ whose time complexity is polynomial in $\rho$.

**Theorem 1.** *The hierarchical key distribution KeyDistribution, based on inner product space, provides stronger security than the key recovery security defined above.*

*Proof*: Let $\mathfrak{g}$ be an adversary that possesses all private information of the child groups for a given group $G$. In this context, let $W$ be the space assigned to $G$, and let $B$ be a basis for $W$, where $\dim W = n$. For simplicity, we assume that $B$ is an orthonormal basis. The adversary $\mathfrak{g}$ also has the corresponding basis for the child groups.

The parent group $G$ has a space $W$ whose dimension is greater than the union of all child group subspaces $W_i$, where each $W_i$ is contained within $W$. For simplicity, we assume that $g$ possesses all basis elements of $B$ except for $b_1$. We know that:

$$K_G = \langle g, \text{Proj}_W f \rangle = \langle g, \sum_{i=1}^{n} b_i \langle f, b_i \rangle \rangle = \underbrace{\langle g, b_1 \rangle \langle f, b_1 \rangle}_{\text{The adversary } \mathfrak{g} \text{ doesn't know}} + \underbrace{\sum_{i=2}^{n} \langle g, b_i \rangle \langle f, b_i \rangle}_{\text{The adversary } \mathfrak{g} \text{ knows}} \qquad (16)$$

For the adversary $\mathfrak{g}$, there are infinitely many possible choices for the vector $b_1$. This is because $W$ is a subspace of a universal space $\Omega$, and the fact that $\dim W$ is strictly smaller than $\dim \Omega$ implies that there exist infinitely many choices for $b_1$. Specifically, there are infinitely many subspaces of $\Omega$ with the same dimension as $W$, each containing the vectors $b_2, \dots, b_n$. $\qquad \square$

The above theorem establishes that colluding child groups cannot derive their parent's basis set. Similarly, same-level groups, as they do not possess the complete basis set, are unable to access each other's keys. Even if multiple adversarial users from child groups $\{G_j, G_k, \dots\}$ pool all their private basis elements, they will always lack at least one critical basis vector necessary to reconstruct $B$, rendering unauthorized key derivation infeasible. The exclusion of the parent group's secret vector ensures security. Consequently, our scheme satisfies $KR_s$ and remains resistant to collusion attacks.

To complement the hardness-based argument above, we introduce a game-based experiment for $KR_s$ in the static setting. Let $Adv_{\text{Stat}_G}^{KR}(1^\rho, \mathfrak{G})$ represent the advantage of a static adversary that corrupts all users in $G_j \in V \setminus \{G \cup \{G_* \mid G_* \leq G\}\}$, acquiring the keys of all subordinate groups. The adversary then attempts to compute the secret basis for $W$ assigned to $G$. In such a scenario, the adversary may reconstruct the secret key $K_G$. However, by design, the union of all child groups' bases forms only a strict subspace $W'$ of the parent space $W$, preventing full recovery of the parent's basis.

An adversary, or multiple collaborating adversaries—seeks to reconstruct their parent's basis and hence to derive the secret key which is computed as the inner product of two vectors, one of which is the projection to the parent's space. The game unfolds as follows: the challenger generates the hierarchy and associated keys,

$$(B, K) \leftarrow \text{Set}(1^\rho, \mathfrak{G}),$$

and the adversary is given $B_j$ for $G_j$ and $K_{G_j}$ for $G \leq G_j$. The adversary then produces an output key $K'$. The game returns 1 if $K' = K_G$, signifying a successful key recovery. The advantage of the adversary is then defined as:

$$\text{Adv}^{KR}_{\text{Stat}_{G_i}}(1^\rho, \text{Grp}) = \text{Pr}[\text{ Adversary successfully outputs } K_G ]. \tag{17}$$

Given that deriving $K_{G_i}$ requires solving the Closest Vector Problem in an Inner Product Space (CVP-IPS) in a higher-dimensional space, while having knowledge of its lower-dimensional subspace. This advantage is negligible due to the following two observation.

**Theorem 2.** *Let $W_1$ be a subspace of the universal space $\mathbb{R}^m$ with a dimension $t_1 < m - 2$. There exist uncountable many subspaces $W$ of $\mathbb{R}^m$ with dimension $t_1 + 1$ such that $W_1 \subset W$.*

*Proof*: Intuitively, this is easy to visualize when $W_1$ is a one-dimensional subspace. In this case, all vectors in $W_1$ lie along a single straight line. It follows that there are uncountably many distinct straight lines, each of which, when combined with the line spanning $W_1$, forms a two-dimensional plane. □

**Theorem 3.** *Let $f$ be a vector in the subspace $W_1$ of the universal space $\mathbb{R}^m$, where $W_1$ has dimension $t_1$ with $t_1 < m-2$. For any $a$, there exist uncountably many vectors $g$ in any subspace $W \supset W_1$ of dimension $t_1 + 1$ such that $\langle f, g \rangle = a$.*

*Proof*: Let $f$ and $a$ be as defined above. Let $W' \supset W_1$ be a subspace of dimension $t_1 + 1$, and let $g \in W'$ be a random vector. Define

$$b = \langle f, g \rangle. \tag{18}$$

Now, consider the vector

$$g' = \frac{a}{b} g. \tag{19}$$

in the subspace $W'$. It is straightforward to observe that

$$\langle g', f \rangle = a. \tag{20}$$

□

The two theorems above establish that even if all child groups act with hostility, collaborate to share their secrets, and manage to obtain partial information about the secret $a$, it remains infeasible for them to infer any details about their parent group's secret. Consequently, our scheme achieves $KR_s$ security under the hardness assumption. In other words, the scheme ensures unconditional security, even in scenarios where all child groups collude.

The problem of deriving a parent group's secret key using only the basis vectors of colluding child groups reduces to solving the projection of a vector in a space without knowing the underlying space. This challenge was previously formalized and studied in [7]. Therefore, the security of our scheme against such attacks is grounded in the computational infeasibility of this problem.

**Strong key indistinguishability - $SKI_s$.** To formally establish indistinguishability, we define two security experiments based on game-based definitions commonly employed in hierarchical key assignment schemes. According to this principle, an attacker should be unable to differentiate between the secret key and a random string of the same length.

For an adversary attempting to distinguish between a random string and the secret vector of a group, they must determine whether a given random vector belongs to the subspace associated with that group. However, by Theorem 1 and 2 for an attacker or a member of any child group, identifying a unique linearly independent vector within the parent group's space is computationally infeasible. This is due to the fact that any element within the universal set, outside the subspace of the child group, has an equal likelihood of being the vector they seek.

Consequently, the probability of correctly identifying a basis vector of the upper group is negligible, rendering the reconstruction of the parent group's basis computationally infeasible. As a result, our scheme satisfies $KI_s$.

Freire et al. [25] presented the concept of strong key indistinguishability $SKI_s$, which implies $KR_s, KI_s$ security models. In this section, we only perform a formal security analysis for $SKI_s$ with a static adversary, as the security models for static and dynamic adversaries are polynomially equivalent [34].

Let's consider a static adversary $Stat_G$, who aims to attack a group $G \in \mathfrak{G}$ and has the ability to

- corrupt all users in each child group $G_j$ of $G$.
- get $K_{G_j}$ for each group $G_j$.

where $i \neq j$, with the mappings $Corrupt_{G_i}$ and $Keys_{G_i}$ respectively. Let's consider the following two experiments [47]:

> Experiment $\mathbf{Exp}_{Stat_G}^{Strong-Ind-1}(1^\rho, \mathfrak{G})$
> $(B, K_G, f, g) \leftarrow Set(1^\rho, \mathfrak{G})$
> $corr_G \leftarrow Corrupt_G(B)$
> $d \leftarrow Stat_G(1^\rho, \mathfrak{G}, f, g, corr_G)$
> return $d$.
> Experiment $\mathbf{Exp}_{Stat_{G_i}}^{Strong-Ind-0}(1^\rho, \mathfrak{G})$
> $(B, K_G, f, g) \leftarrow Set(1^\rho, \mathfrak{G})$
> $corr_G \leftarrow Corrupt_G(B)$
> $T \xleftarrow{r} \{0, 1\}^\rho$
> $d \leftarrow Stat_G(1^\rho, \mathfrak{G}, f, g, corr_G, T)$
> return $d$.

It is the responsibility of $Stat_G$ to determine whether the challenge received corresponds to $K_G$ or $T$. The advantage of $Stat_G$ can be defined as

$$\text{Adv}_{Stat_G}^{\text{Strong-Ind}}(1^\rho, \mathfrak{G}) = \left| \Pr\left[ \text{Exp}_{Stat_G}^{\text{Strong-Ind-1}}(1^\rho, \mathfrak{G}) = 1 \right] - \Pr\left[ \text{Exp}_{Stat_G}^{\text{Strong-Ind-0}}(1^\rho, \mathfrak{G}) = 1 \right] \right|. \tag{21}$$

A scheme is considered secure in the sense of strong key indistinguishability against static adversaries if $\text{Adv}_{Stat_G}^{\text{Strong-Ind}}(1^\rho, \mathfrak{G})$ is negligible for each $\mathfrak{G} = (V, E) \in \Gamma$. This probability quantifies the likelihood of an adversary distinguishing a secret key from a random string of the same length.

The following theorem establishes that this probability remains negligible, meaning that distinguishing between the actual secret and a random string is independent of any computational problem, even for an adversary possessing all information from child groups.

**Theorem 4.** *Let $\mathfrak{g}$ be an adversary with access to all secrets of the child groups of G. Let d be an arbitrary string of the same length as $K_G$, the secret key constructed by the members of G. Then, determining whether $d = K_G$ is computationally infeasible.*

*Proof*: Let $W$ be the subspace associated with group $G$. The union of all child groups' spaces forms a strict subspace of $W$, meaning that even if all child groups collaborate and share their basis elements, they will still lack at least one vector necessary to form a complete basis for $W$.

Now, let $W' \subset W$ represent the adversary's subspace, and let $f, g$ be public vectors. The secret key generated by group $G$ is given by:

$$K_G = \langle \text{Proj}_W f, g \rangle. \tag{22}$$

According to Theorem 2, the adversary has infinitely many possible choices for spaces $W''$ of dimension $\dim W' + 1$, such that $W' \subset W''$. This implies that even though $W$ has a higher dimension than $W'$, the adversary still faces an infinite number of possibilities when attempting to determine the complete space $W$.

In other words, the missing basis elements introduce uncertainty, preventing the adversary from reconstructing $W$ completely, even if $W$ maybe only slightly larger than $W'$ (i.e., $\dim W = \dim W' + 1$). As a result, the adversary has infinitely many choices for $\text{Proj}_W f$ when only knowing $\text{Proj}_{W'} f$, which in turn leads to infinitely many possibilities for the final secret key $K_G$. $\qquad\square$

The resulting keys are derived solely from the knowledge of a basis for the subspace assigned to a group. Theorems 2 and 3 establish that even possessing $t$–1 basis elements of a space with dimension $t$ provides no information about the space itself, as every possible space of dimension $t$ containing these $t$–1 elements has an equal probability of being $W$. Consequently, the probability of accurately determining whether the result of an inner product is obtained using the basis of the correct subspace is negligible. Therefore, the proposed scheme is secure under $SKI_s$.

**Resistance to the privilege creep problem.** In a hierarchical access control system, privilege creep occurs when users retain access to groups or resources they should no longer be authorized to access due to structural changes in the organization. To prevent this, a secure key assignment scheme must ensure that revoked users or downgraded groups cannot derive new keys after an update.

- *Adversary Model:* Consider an adversary $\mathcal{A}$, who was initially a member of group $G_i$ and had access to the corresponding basis set $B_i$. Due to a structural change—such as a classification level downgrade of $G_i$ or a membership modification (e.g., a user $u_{ij}$ moving to another group)—the adversary loses legitimate access to certain group keys. The goal of $\mathcal{A}$ is to leverage previously obtained keys, basis sets, or any available public information to derive the new key for $G_i$ or any related groups.
- *Security Mechanism:* To prevent $\mathcal{A}$ from retaining unauthorized access, the system enforces key revocation and re-randomization upon structural updates. Specifically:
  – If $G_i$ undergoes any modification, its basis set $B_i$ is recomputed and updated.
  – Algorithm 1 is executed by the data owner from step 8, recursively updating $G_i$ and all its child groups.
  – The previous basis sets are overridden and permanently revoked, ensuring that they are not transferable to newly added groups.
  – Since the new basis set is randomly re-selected from a high-dimensional vector space, it is computationally infeasible for a revoked user to reconstruct or guess the new basis.
- *Security Implication:* Even if $\mathcal{A}$ retains historical basis sets or previously obtained keys, they lack the necessary information to compute updated keys. Without knowledge of the new secret basis vector, the adversary must solve a randomized linear system, which is an underdetermined problem and thus infeasible in polynomial time. Consequently, our scheme effectively mitigates the privilege creep problem, ensuring that revoked users cannot maintain unauthorized access.

**Secure rekeying process and trust model.** The scheme does not rely on an external TTP. The data controller acts as the trust anchor. New basis sets are securely distributed using standard cryptographic channels (e.g., TLS, secure VPN, or hardware-based enclaves). To minimize exposure, ephemeral keys may be used during distribution, and key confirmation protocols (e.g., challenge–response) can be employed to verify correct delivery without disclosing basis content. This approach ensures that revoked users cannot reuse old credentials while keeping system availability intact.

**Remark 3.** *In hierarchical key assignment systems, the detection of compromised group bases must occur at the operational layer. The cryptographic scheme itself lacks the inherent capability to determine whether an assigned basis has*

*been compromised; if such a compromise occurs, an adversary can trivially derive the current key. Indicators of unauthorized activity such as anomalous key derivations or irregular access attempts, can instead be identified through systematic access-log monitoring. Upon confirmation of compromise, the selective rekeying mechanism outlined in Algorithm 2 is initiated. This mechanism recomputes and securely redistributes only the affected subtree, thereby minimizing communication overhead while preserving the integrity and continuity of the remaining hierarchy.*

**Quantum resistance.** While the previous subsections demonstrate our scheme's robustness against classical adversaries, it is also essential to evaluate its security posture in the context of quantum-capable threat actors.

Unlike traditional hierarchical key assignment systems that rely on number-theoretic assumptions—such as RSA, ECC, or Diffie–Hellman—which are vulnerable to Shor's algorithm due to their dependence on integer factorization or discrete logarithms, our construction is inherently immune to such weaknesses. The proposed scheme is built upon randomized subset selection, basis exclusion, and linear operations in high-dimensional vector spaces, which do not exhibit the algebraic structure required for quantum algorithm exploitation. Specifically:

- **Shor's algorithm:** Our scheme avoids primitives based on factorization or discrete logarithm assumptions. Since Shor's algorithm is designed to break such structures efficiently on a quantum computer, it has no applicability to our construction.
- **Grover's algorithm:** Although Grover's algorithm offers a quadratic speedup for brute-force search, the key generation process in our scheme depends on randomly selecting subsets from random subsets within the universal space $\mathbb{R}^m$. As Theorems 2 and 3 suggest, there are infinitely many choices for the subspaces being searched. Consequently, using brute-force random search for such spaces is not feasible.
- **Lattice-based quantum attacks:** Our system does not rely on problems such as Learning With Errors (LWE), Closest Vector Problem (CVP), or Shortest Vector Problem (SVP), which form the foundation of lattice-based cryptography. Therefore, it is not susceptible to classical or quantum-accelerated lattice reduction attacks, including BKZ, sieving, or quantum walk algorithms. Although our key recovery security is formally reduced to a CVP-like problem in an inner product space (CVP-IPS), this reduction is used solely for security hardness justification and should not be confused with lattice-based constructions, which exhibit entirely different structural assumptions and are directly targeted by quantum-accelerated lattice reduction attacks.

For applications that require formal post-quantum assurances or compliance with regulatory standards, our design can be optionally hybridized with NIST-recommended post-quantum Key Encapsulation Mechanisms (KEMs), such as CRYSTALS-Kyber. In this hybrid model, the group keys or basis vectors are encrypted using post-quantum secure primitives, providing an additional confidentiality layer without altering the logic or structure of the hierarchical key assignment.

By decoupling its security from number-theoretic and lattice-based assumptions and enabling optional integration with PQC standards, the proposed scheme offers a modular cryptographic framework for deployment in both classical and post-quantum environments.

**Parameter recommendations based on NIST security levels.** The security of the scheme depends primarily on three parameters assuming the universal space is $\mathbb{R}^m$:

- the universal space dimension $m$,
- the dimension $n_i$ of $W_i$, and
- the bit-length $\lambda$ of each coordinate of the basis elements.

To align the scheme with the NIST PQC security categories (Levels 1, 3, and 5) and the key-strength guidelines defined in NIST SP 800-57, we recommend the following parameter sets:

NIST Level 1 (128-bit classical security). A minimum configuration ensuring $2^{128}$ operations for brute-force reconstruction:

$$m \geq 16, \quad n_i \geq 5, \quad \lambda = 32. \tag{23}$$

NIST Level 3 (192-bit classical security). A stronger configuration preventing any adversary below $2^{192}$ operations:

$$m \geq 32, \quad n_i \geq 5, \quad \lambda = 64. \tag{24}$$

NIST Level 5 (256-bit classical security). Full-strength configuration consistent with AES-256 resistance:

$$m \geq 64, \quad n_i \geq 5, \quad \lambda = 128. \tag{25}$$

These recommendations ensure that any attempt to reconstruct the protected subspace or distinguish the group key from uniform requires computational effort consistent with the NIST PQC security categories. Importantly, these parameter sets do not alter the hierarchical structure or the operational behavior of the scheme; only the vector dimensions and coordinate sizes are scaled according to the desired security level.

## Comparison with other schemes

Table 4 provides a detailed comparison of our key access scheme with several widely recognized schemes. The evaluation considers various parameters critical for assessing the efficiency, security, and practicality of these schemes. The notations used for these parameters are explained at the bottom of Table 4 for clarity and consistency.

Several Akl and Taylor-based schemes, as reviewed in the "Literature review" section [2,9]–[16,19,20], involve computationally expensive key update and derivation processes. As a result, these schemes may not be practical for hierarchical structures with a large number of distinct groups (classes). Additionally, some of these schemes are susceptible to collusion attacks and achieve $KR_s$ and $KI_s$ only under the RSA assumption or the random exponent assumption. It is worth noting that the provably secure scheme, which is both $KR_s$ and $KI_s$, was first introduced by Atallah [18].

In previous works such as [18,32], the amount of public information increases as the number of security classes and edges in the graph increases. Each class only requires a single symmetric key for private storage. However, the time cost of key derivation is directly proportional to the path length between classes, which means that the time-cost increases as the path length increases. These schemes are considered $KI_s$ and chosen-plaintext-attack-resistant $CPA_r$ from a security standpoint.

According to [17], the Akl and Taylor-based schemes [9,16,19,20] are proven to be $KI_s$ only under the RSA assertion ($RSA_s$). However, to satisfy the additional requirements of public storage and key derivation time for $KI_s$, the cost will be several times greater than the bit length $l$ of the key.

The key derivation process in [21] is highly efficient, even for cases where the path length between classes is significantly large. However, the scheme has notable drawbacks in terms of storage requirements. In the first scheme, the public storage requirement depends on both the number of time periods and classes, while in the second scheme, the private storage requirement is determined primarily by the number of time periods.

In [25], a trade-off exists between the need for private information storage and the efficiency of key derivation, which is constrained by the path length $h$ between classes in the hierarchical structure. Additionally, the amount of private information required is determined by the poset width $w$. A notable contribution of this scheme is that it eliminates the need for public storage, making it a significant advancement in the literature.

The paper [34] proposes a direct access scheme, which is denoted as $SKI_s$, that avoids the need for iterative computation and handles potential hierarchical changes with minimum computations. However, one major drawback of this

**Table 4**. Comparison with previous schemes.

| Schemes | D | PriS | PubS | KD | ST | SA |
|---|---|---|---|---|---|---|
| Akl and Taylor based [2], [9]–[16,19,20] | No | One | $c.t$ | $t_{exp}$ | $KR_s$ | N/A |
| Atallah et al. [18] | Yes | One | $e + c$ | $h.(DT_{se} + T_{prf})$ | $KI_s$ | $CPA_s + PRF$ |
| De Santis et al. [32] | Yes | One | $e + 2c$ | $(h + 2).DT_{se}$ | $KI_s$ | $CPA_s$ |
| D'Arco et al. [17] | Yes | One | $l.c$ (min) | $l.DT_{se}$ | $KI_s$ | $RSA_s$ |
| Ateniese et al. TLEBC [21] | Yes | One | $c^2.t^3$ (max) | $DT_{se}$ | $KI_s$ | $CPA_s$ |
| Ateniese et al. TLPBC [21] | Yes | $t$ (max) | $c^2$ (max) | $DT_p$ | $KI_s$ | $CPA_s$ |
| Freire et al. PRF-based [25] | No | $w$ | Zero | $h.T_{prf}$ | $SKI_s$ | PRF |
| Freire et al. FSPRG-based [25] | No | $w$ | Zero | $h.T_{prg}$ | $SKI_s$ | FSPRG |
| Tang et al. [34] | Yes | Four | $c^2 + 1$ (min) | $2A + 4M$ | $SKI_s$ | PRF |
| Celiktas et al. [36] | Yes | One (min), $c$ (max) | Zero | $d^2.\log p$ | $KI_s$ | TSBITS |
| Celikbilek et al. [8,48,49] | Yes | $\sigma_1$ | Two | $\sigma_2$ | $SKI_s$ | CVP–IPS |
| The proposed scheme | Yes | s+1(max) | Two | $OP_t$ | $SKI_s$ | LICV |

**Notation:**

$D$ : Dynamic

$Pri_S$ : Private Storage

$Pub_S$ : Public Storage

$KD$ : Key Derivation

$ST$ : Security Type

$SA$ : Security Assertions

$c$ : Number of groups (classes)

$t$ : Number of time periods

$t_{exp}$: Exponential time over a large group/class ($D + E$).

$KR_s$ : Key recovery secure

$N/A$ : Not applicable

$e$: Number of edges

$h$ : Path length between the groups (classes)

$DT_{se}$ : Decryption time using a symmetric key encryption scheme

$T_{prf}$ : Time of PRF evaluation

$KI_s$ : Key indistinguishability secure

$CPA_s$ : Chosen-plaintext-attack-resistant

$PRF$ : Pseudorandom function

$l$ : Bit length of the key

$RSA_s$ : Secure under the RSA assumption

$DT_p$ : Decryption time using pairing evaluation

$w$ : Poset width

$SKI_s$ : Strong key indistinguishability secure

$T_{prg}$ : Time of PRG evaluation

$FSPRG$ : Forward-secure pseudorandom generator

$A$ : Computational time of modular addition over $F_q$

$M$ : Computational time of modular multiplication over $F_q$

$d$ : Interpolation polynomial degree

$p$ : Number of elements in $F_p$

$TSBITS$ : Threshold scheme-based information-theoretic secure

$s$ : Product of number of groups in the hierarchy and maximum number of members in a group.

$OP_t$ : Time-cost using orthogonal projection

$LICV$ : Linearly independent chosen vectors

$\sigma_1$ : Represents the sum of the basis set length and the number of descendant groups, see [8,48,49].

$\sigma_2$ : Corresponds to the time taken for orthogonalization and orthogonal projection procedures, as defined in [8,48,49].

$CVP–IPS$ : Closest vector problem defined in an inner product space.

scheme is its public storage requirement, which is higher than other schemes. The storage need has a linear relation with computational cost. To derive the key for each class, two modular multiplication costs $M$ and one modular addition cost $A$ over $F_q$ need to be performed. Furthermore, each class requires the calculation of $2A$ and $4M$ over $F_q$. Consequently, the

computational cost for each class could become prohibitive, rendering the scheme inefficient. Additionally, updating the hierarchy necessitates that the data controller compute and publish a new public matrix, further increasing the operational complexity.

In [36], the storage need for private information is proportional to the number of classes $G$ in a specific organizational unit (OU). However, it can be reduced to just one class, whereas the storage need for public information is zero. Moreover, the time cost for key derivation is quadratic and equal to the square of the employed polynomial degree.

In [8], Celikbilek et al. formally defined the CVP-IPS problem and introduced a generic hierarchical key assignment scheme (HKAS) based on it. This generic scheme was subsequently implemented within a cloud computing environment in [48]. A comprehensive synthesis and detailed exposition of these works are provided in [49]. The scheme's public storage requirement is two, demonstrating efficient use of publicly accessible resources. The scheme provides strong key indistinguishability $SKI_s$ and is based on the CVP-IPS as its security foundation.

On the other hand, our proposed scheme provides a flexible and fine-grained hierarchical direct key assignment and access control mechanism, eliminating the need for other groups or classes during key derivation. Unlike existing hierarchical key assignment schemes, our approach minimizes public storage while maintaining efficient key derivation. The key derivation complexity remains at $\mathcal{O}(n_j^2)$, where $n_j$ represents the dimension of the corresponding vector space. Experimental results confirm that the scheme meets all desirable performance criteria, ensuring efficiency even in larger hierarchies.

The system-ready time cost remains within acceptable limits, with key derivation time scaling directly with the number of classes, making it practical for real-world applications. A key advantage of our scheme over Tang et al. [34] is that it does not require the publication of new public information when hierarchical changes occur, thereby reducing computational overhead and maintaining privacy. Furthermore, our scheme supports dynamic modifications while ensuring that storage and computational costs remain independent of the number of updates.

From a security perspective, we have formally proven that our scheme guarantees strong key indistinguishability security ($SKI_s$), offering superior protection against key inference attacks compared to earlier schemes. Unlike schemes that rely on pseudorandom functions (PRF) or exponentiation-based key derivation, our approach leverages linearly independent chosen vectors (LICV), ensuring resistance to collusion attacks and privilege creep.

Compared to state-of-the-art hierarchical key assignment schemes, our approach achieves a better balance between efficiency and security, reducing public storage requirements, optimizing key derivation complexity, and eliminating costly cryptographic dependencies. The comparative analysis (Table 4) demonstrates that our scheme outperforms traditional methods by minimizing storage, reducing key update costs, and ensuring cryptographic robustness. Consequently, our scheme is highly scalable, computationally efficient, and well-suited for modern access control environments.

## Conclusion and future work

This work leverages inner product spaces to design a hierarchical key assignment scheme suitable for various applications, particularly cloud-based environments and large-scale access control systems.

To evaluate our scheme against existing hierarchical key assignment and access control mechanisms, we considered key metrics such as private and public storage requirements, key derivation complexity, key update efficiency, and compliance with security properties including $KR_s$, $KI_s$, $SKI_s$, and resistance to the privilege creep problem. One of the major limitations in hierarchical infrastructures is the excessive storage demand for private and public information, which imposes a significant burden on data controllers handling mission-critical data. Our scheme effectively mitigates this issue by reducing these storage requirements to practical levels while maintaining flexibility and scalability.

A critical advantage of our scheme is its computational efficiency in both key derivation and updates. Unlike traditional schemes that rely on complex cryptographic primitives or require frequent re-keying, our design optimizes key derivation with a reduced number of vectors, leading to lower computational overhead. This efficiency directly translates into improved performance, particularly in large hierarchical structures.

To ensure secure and unpredictable subset selection during hierarchical updates, our scheme utilizes a cryptographically secure pseudorandom number generator (CSPRNG). The generator provides randomness that is used to select a subset of basis vectors, thereby preserving the linear independence and entropy of the updated user space. The CSPRNG is seeded during the selection of random coefficients $c_i$, which are used to construct the vectors $v_j = c_1 w_1 + \cdots + c_n w_n$ for $j<n$, where $\{w_1, \dots, w_n\}$ represents the basis of our highest group space.

Compared to state-of-the-art hierarchical key assignment schemes, our approach achieves a better balance between efficiency and security. The comparative analysis (Table 4) demonstrates that our scheme outperforms traditional methods by minimizing storage, reducing key update costs, and ensuring cryptographic robustness.

From a security perspective, the scheme effectively addresses major concerns such as collusion attacks and unauthorized access due to privilege creep. Our approach ensures robust resistance against such threats, providing both forward and backward security, and fully satisfying the $KR_s$, $KI_s$, and $SKI_s$ security properties. Furthermore, our security analysis confirms the scheme's resilience under strong adversarial models, reinforcing its suitability for dynamic and evolving access control environments.

While the security claims are predominantly asymptotic, we emphasize that for practical parameter selections (e.g., vector dimension $m = 128$), recovering the root basis from compromised user vectors equates to solving an underdetermined system with an infinite search space. For instance, the brute-force effort required to identify a parent's subspace within a 128-dimensional space is infeasible, as there exist infinitely many potential candidate subspaces. Another important concern is the potential leakage of basis vectors in the event of partial user compromise. Our scheme demonstrates resilience even if multiple user-specific bases $B_{ij}$ are compromised in such scenarios. Each $B_{ij}$ is generated using random coefficients applied to the root basis $B_1$, without disclosing the original vectors or transformation mappings. Consequently, reconstructing $B_1$ from multiple $B_{ij}$ vectors is equivalent to identifying the parent space using only vectors that span a strict subspace of it—an infeasible task regardless of the adversary's computational power.

We also recognize the cryptographic need for precise key derivation unaffected by numerical imprecision. While the scheme was implemented over a real-valued vector space over $\mathbb{R}$ for computational simplicity and theoretical clarity, we acknowledge the importance of avoiding floating-point errors in cryptographic systems. Therefore, a promising direction for future work is to transition the scheme to a lattice-based model with discrete vector representations and quantized key outputs, thus enhancing compatibility with practical cryptographic requirements.

Moreover, the current design's reliance on continuous vector spaces avoids the vulnerabilities associated with certain computationally hard problems found in traditional public key algorithms. However, while integrating discrete inner product structures could offer advantages, such as mitigating any potential instability during operations, it may also introduce vulnerabilities. Specifically, as seen in lattice-based systems, this integration could reduce the scheme's security to a computationally solvable problem.

In addition to theoretical and efficiency considerations, practical deployment scenarios also merit discussion. The proposed scheme can be integrated with existing PKI and IAM infrastructures. Derived group keys $K_{G_i}$ can be used as symmetric session keys (e.g., AES-256) or embedded in certificate structures for secure communication. Such compatibility allows the scheme to serve as a backend key management layer while interfacing with standardized identity and access management systems. Future work may explore formal APIs and wrappers that enable such seamless integration.

Additionally, while the current security proofs rely on well-established game-based arguments (e.g., $SKI_s$), future work may consider formalizing the scheme within the Universal Composability (UC) framework. Doing so would provide stronger composability guarantees and enable seamless integration with cryptographic protocols that require UC-security, such as secure file sharing or key exchange systems.

Furthermore, enhancing fault tolerance by enabling secure regeneration of corrupted user-specific bases $B_{ij}$, either via deterministic derivation from securely stored seeds or by distributing the root basis $B_1$ through a secret-sharing mechanism, is a promising direction for robust deployment.

Although our security proofs address mathematical and structural resilience, practical deployments of the scheme must consider side-channel resistance. In particular, the GS orthogonalization and inner product computations are susceptible to timing and power-based side-channel attacks due to their operand-sensitive control flow and memory access patterns. To mitigate such threats, implementations should adopt constant-time algorithms, masking techniques, and side-channel-hardened libraries. Future work will explore formal side-channel leakage models and implement the scheme using side-channel resistant primitives to ensure robustness against practical exploitation.

To ensure bit-exact key derivation and eliminate inconsistencies due to machine-dependent rounding behavior, a future variant of the scheme may consider discrete vector spaces, such as integer lattices ($\mathbb{Z}^m$) or finite fields ($\mathbb{F}_p^m$), where exact arithmetic is feasible. This approach not only improves computational determinism but also facilitates verifiable implementations and formal correctness proofs in cryptographic settings.

Overall, the proposed scheme achieves a balanced trade-off between security, efficiency, and scalability, making it a practical and robust solution for hierarchical key assignment across both established infrastructures and emerging application domains, particularly those requiring dynamic structure, scalability, and post-quantum resilience in key management.

## Supporting information

**S1 Appendix. Source Code for the Hierarchical Key Assignment Scheme.**
(HTML)

## Author contributions

**Formal analysis:** Enver Ozdemir.

**Methodology:** Baris Celiktas.

**Software:** Baris Celiktas, Ibrahim Çelikbilek.

**Supervision:** Enver Ozdemir.

**Validation:** Baris Celiktas, Ibrahim Çelikbilek, Sueda Guzey.

**Visualization:** Baris Celiktas.

**Writing – original draft:** Baris Celiktas, Ibrahim Çelikbilek, Sueda Guzey, Enver Ozdemir.

**Writing – review & editing:** Baris Celiktas, Enver Ozdemir.

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
