## [Decision Letter · Decision Letter 0]

27 Nov 2025

PONE-D-25-38934Hierarchical Secure Key Assignment SchemePLOS ONE

Dear Dr. Celiktas,

Thank you for submitting your manuscript to PLOS ONE. After careful consideration, we feel that it has merit but does not fully meet PLOS ONE’s publication criteria as it currently stands. Therefore, we invite you to submit a revised version of the manuscript that addresses the points raised during the review process. Please submit your revised manuscript by Jan 11 2026 11:59PM. If you will need more time than this to complete your revisions, please reply to this message or contact the journal office at plosone@plos.org. Please include the following items when submitting your revised manuscript:

We look forward to receiving your revised manuscript.

Kind regards,

Sohail Saif, Ph.D

Academic Editor

PLOS ONE

Journal Requirements:

Reviewers' comments:

Reviewer's Responses to Questions

**Comments to the Author**

1. Is the manuscript technically sound, and do the data support the conclusions?

Reviewer #1: Yes

Reviewer #2: Yes

Reviewer #3: Yes

2. Has the statistical analysis been performed appropriately and rigorously?

Reviewer #1: N/A

Reviewer #2: Yes

Reviewer #3: Yes

3. Have the authors made all data underlying the findings in their manuscript fully available?

Reviewer #1: Yes

Reviewer #2: Yes

Reviewer #3: No

4. Is the manuscript presented in an intelligible fashion and written in standard English?

Reviewer #1: Yes

Reviewer #2: Yes

Reviewer #3: Yes

5. Review Comments to the Author

Reviewer #1: 1) The authors presented a novel hierarchical key assignment mechanism for access control, designed to be computationally lightweight and optimized for digital environments with structured access policies by using orthogonal projection and distributing a basis to each group.

2) The ideal is uique, but the same idea has already been published in the following IEEE Access article "A Hierarchical Key Assignment Scheme: A Unified Approach for Scalability and Efficiency" with the same authors. The article submitted in PLOS ONE article appears to be a continuation or extension of the plan presented in the IEEE Access article, with more extensive contributions but overlapping authors and methodology, and cited as reference 42 in this paper.

3) The authors presented a detailed complexity analysis of the algorithm along with benchmark with the existing work. However, equations number are not properly mentioned, and equation number style and citation number style are bot same ().

4) Figure 4 needs to be redesign.

Reviewer #2: 1- The abstract did not mention any numerical values for the metrics that demonstrate the superiority of the proposed technology.

2- It would be best to include a table summarizing the advantages and disadvantages of each research paper included in the related works to highlight what has been overcome through this submitted manuscript.

3- It would have been better to present the proposed algorithm in Figure 4 in a more detailed way, rather than just including explanatory text without any visual illustrations.

4- A large proportion of the references were published before 2020, so it is preferable to cover all of the last five years, especially the period between 2023-2025, with no less than eight research papers within the last three years.

Reviewer #3: A study presented novel ideas based on orthogonal projection and distributing a basis. However paper should be accepted after minor revisions and corrected the following issues.

1.Add more recently published works

2.Clearly present your contributions compared with recent related works in table form.

3.Explain how does the system detect if a basis set has been compromised?

4. Add comprehensive benchmarks including memory usage, network bandwidth for key distribution.

5.Add a concrete security analysis section with parameter recommendations aligned to NIST security levels.

6. PLOS authors have the option to publish the peer review history of their article (what does this mean?). If published, this will include your full peer review and any attached files.

Reviewer #1: No

Reviewer #2: No

Reviewer #3: No

---

## [Author Response · Author response to Decision Letter 1]

18 Dec 2025

Dear Editor,

All requested revisions have been completed and the detailed “Response to Reviewers” document has been uploaded as a PDF. Please let me know if any additional information is required.

Kind regards.

---

## [Decision Letter · Decision Letter 1]

11 Jan 2026

Hierarchical Secure Key Assignment Scheme

PONE-D-25-38934R1

Dear Dr. Celiktas,

We’re pleased to inform you that your manuscript has been judged scientifically suitable for publication and will be formally accepted for publication once it meets all outstanding technical requirements.

Kind regards,

Sohail Saif, Ph.D

Academic Editor

PLOS One

Additional Editor Comments (optional):

Reviewers' comments:

Reviewer's Responses to Questions

**Comments to the Author**

1. If the authors have adequately addressed your comments raised in a previous round of review and you feel that this manuscript is now acceptable for publication, you may indicate that here to bypass the “Comments to the Author” section, enter your conflict of interest statement in the “Confidential to Editor” section, and submit your "Accept" recommendation.

Reviewer #1: All comments have been addressed

Reviewer #2: (No Response)

Reviewer #3: All comments have been addressed

2. Is the manuscript technically sound, and do the data support the conclusions?

Reviewer #1: Yes

Reviewer #2: (No Response)

Reviewer #3: Partly

3. Has the statistical analysis been performed appropriately and rigorously?

Reviewer #1: Yes

Reviewer #2: (No Response)

Reviewer #3: Yes

4. Have the authors made all data underlying the findings in their manuscript fully available?

Reviewer #1: No

Reviewer #2: (No Response)

Reviewer #3: (No Response)

5. Is the manuscript presented in an intelligible fashion and written in standard English?

Reviewer #1: Yes

Reviewer #2: (No Response)

Reviewer #3: Yes

6. Review Comments to the Author

Reviewer #1: All comments have been addressed by the authors thoroughly. I think this manuscript is now suitable to publish in this journal.

Reviewer #2: (No Response)

Reviewer #3: (No Response)

7. PLOS authors have the option to publish the peer review history of their article (what does this mean?). If published, this will include your full peer review and any attached files.

Reviewer #1: No

Reviewer #2: No

Reviewer #3: No

---

## [Editor Report · Acceptance letter]

PONE-D-25-38934R1

PLOS One

Dear Dr. Celiktas,

I'm pleased to inform you that your manuscript has been deemed suitable for publication in PLOS One. Congratulations! Your manuscript is now being handed over to our production team.

Kind regards,

on behalf of

Dr. Sohail Saif

Academic Editor

PLOS One